# Learning Mask-aware CLIP Representations for Zero-Shot Segmentation

**Siyu Jiao**[1,2,3,*], **Yunchao Wei**[1,2,3], **Yaowei Wang**[3], **Yao Zhao**[1,2,3], **Humphrey Shi** [4,5]

[1] Institute of Information Science, Beijing Jiaotong University
[2] Peng Cheng Laboratory
[3] Beijing Key Laboratory of Advanced Information Science and Network
[4] Georgia Institute of Technology    [5] Picsart AI Research
jiaosiyu99@bjtu.edu.cn

## Abstract

Recently, pre-trained vision-language models have been increasingly used to tackle the challenging zero-shot segmentation task. Typical solutions follow the paradigm of first generating mask proposals and then adopting CLIP to classify them. To maintain the CLIP's zero-shot transferability, previous practices favour to freeze CLIP during training. However, in the paper, we reveal that CLIP is insensitive to different mask proposals and tends to produce similar predictions for various mask proposals of the same image. This insensitivity results in numerous false positives when classifying mask proposals. This issue mainly relates to the fact that CLIP is trained with image-level supervision. To alleviate this issue, we propose a simple yet effective method, named Mask-aware Fine-tuning (MAFT). Specifically, Image-Proposals CLIP Encoder (IP-CLIP Encoder) is proposed to handle arbitrary numbers of image and mask proposals simultaneously. Then, *mask-aware loss* and *self-distillation loss* are designed to fine-tune IP-CLIP Encoder, ensuring CLIP is responsive to different mask proposals while not sacrificing transferability. In this way, mask-aware representations can be easily learned to make the true positives stand out. Notably, our solution can seamlessly plug into most existing methods without introducing any new parameters during the fine-tuning process. We conduct extensive experiments on the popular zero-shot benchmarks. With MAFT, the performance of the state-of-the-art methods is promoted by a large margin: 50.4% (+ 8.2%) on COCO, 81.8% (+ 3.2%) on Pascal-VOC, and 8.7% (+4.3%) on ADE20K in terms of mIoU for unseen classes. Code is available at github.com/jiaosiyu1999/MAFT.git.

## 1 Introduction

Semantic segmentation, one of the most widely researched topics in computer vision, has achieved remarkable success [3; 39; 15; 16] with the development of deep learning techniques [14]. However, traditional segmentation models are only capable of segmenting a few predefined categories within a closed vocabulary [7; 2; 22; 21], which is much smaller than the number of categories used by humans to describe the real world. Therefore, zero-shot segmentation [31; 1; 10; 13] is introduced to segment objects using arbitrary categories described by texts.

Recently, large-scale visual-language pre-training models (*e.g.* CLIP [28] and ALIGN [17]) have shown impressive transferability in recognizing novel categories, leading to their increased adoption

---

*Work done during an internship at Picsart AI Research (PAIR).

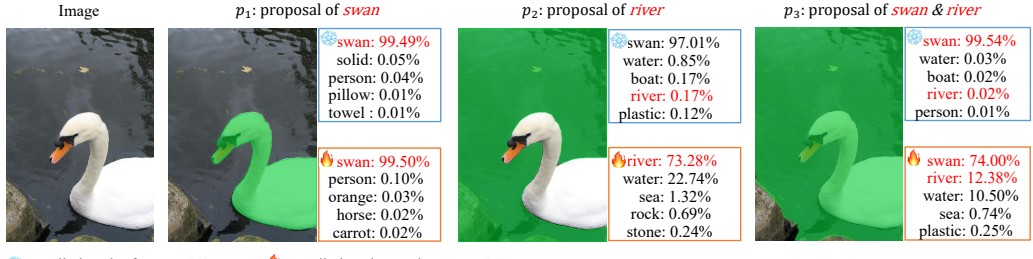

Figure 1: Comparison between the frozen CLIP and our mask-aware CLIP for proposal classification. Regions of proposals are highlighted with green. The frozen CLIP classifies $p_1$, $p_2$, and $p_3$ as *swan* class and produces similar predictions, although these proposals contain different regions of the image. After the MAFT, the mask-aware CLIP can produce proper scores for different proposals.

for tackling the challenging zero-shot segmentation task [6; 33; 20; 27]. A mainstream solution follows the "frozen CLIP" paradigm, which executes the zero-shot segmentation with two steps: 1) first employing a Proposal Generator to produce class-agnostic mask proposals and 2) then leveraging a frozen pre-trained CLIP to classify each mask proposal via similarity matching in the aligned image-text feature space. While acceptable results are obtained, we reveal that these approaches overlook a crucial issue, *i.e.* the frozen CLIP is insensitive to different mask proposals and tends to produce similar predictions for various proposals of the same image.

To better illustrate the above-mentioned issue, we show several examples in Fig. 1. We use Mask-Former [5] to generate a series of mask proposals and select three typical ones. When using frozen CLIP for classification, we observe that it correctly classifies the high-quality *swan* proposal $p_1$. However, for the other two proposals $p_2$ and $p_3$, which respectively contain only shape information of *swan* and both regions of *swan* and *river*, the frozen CLIP produces similar predictions compared to $p_1$. This is reasonable since CLIP is trained by image-text pairs, making it insensitive to pixel-level information (*e.g.* background noise), and resulting in numerous false positives. Based on the above observations, we consider that an expected CLIP for zero-shot segmentation task should **1) be sensitive to different mask proposals, 2) not compromise its original transferability on novel classes.**

To this end, we introduce a Mask-aware CLIP Fine-tuning method (dubbed MAFT). To make CLIP sensitive to different mask proposals, we devise an Image-Proposals CLIP Encoder (IP-CLIP Encoder), which utilizes mask proposals to perform masked Multihead Attention [5; 4]. This design enables the model to handle arbitrary numbers of images and proposals simultaneously. The *mask-aware loss* is proposed to minimise the distance between the IoU score of mask proposals and the classification score of IP-CLIP Encoder, prompting IP-CLIP Encoder to differentiate various proposals. Besides, to preserve CLIP's zero-shot transferability, we utilize a frozen CLIP as a teacher network to facilitate fine-tuning. This is achieved by aligning the outputs of the frozen CLIP and IP-CLIP Encoder through *self-distillation loss*. By performing MAFT, several advantages are provided: 1) Fine-tuning is efficient since only a few mask proposals need to be classified. 2) Compared to pixel-level fine-tuning, mask-aware fine-tuning hardly alters the structure of CLIP itself, preserving its maximum transferability. 3) Mask-aware fine-tuning of CLIP is released from the segmentation module, making it plug-and-play and applicable to any "frozen CLIP" approaches. As shown in Fig. 1, the mask-aware CLIP can well distinguish different proposals and provide proper classification scores for both seen (*river*) and unseen (*swan*) classes.

We evaluate our MAFT on three commonly used zero-shot segmentation benchmarks: COCO-Stuff [2], Pascal-VOC [7], and ADE20K [40]. Extensive experiments show that MAFT works well with various zero-shot segmentation methods. In particular, by plugging MAFT, the state-of-the-art approach FreeSeg [27] achieves superior performance on COCO-Stuff ($42.2\% \rightarrow 50.4\%$), Pascal-VOC ($78.6\% \rightarrow 81.8\%$) and ADE20K ($4.4\% \rightarrow 8.7\%$) in terms of mIoU of unseen classes. Furthermore, we conduct experiments in a *open-vocabulary* setting, where MAFT enhances the performance of A-847 [40], A-150 [40], PC-459 [24], PC-59 [24] and PAS-20 [7] datasets by +3.0%, +11.2%, +6.4%, +19.1% and +4.4%, respectively. Notably, our approach outperforms the freezing CLIP counterpart and establishes new state-of-the-art results on all datasets.

## 2 Related Work

**Zero-Shot Segmentation** [29] is established to break the restriction of categories and perform segmentation on unseen classes. Earlier works SPNet [31] learn a joint pixel and vocabulary concept embedding space, ZS5 [1] utilizes a generative model to generate pixel-level features based on word embeddings of unseen classes, CaGNet [10] incorporates context information for better feature generation. Recent approaches take the advent of large-scale visual-language models (*e.g.* CLIP [28] and ALIGN [17]) to leverage rich alignment features from image-text pairs. [34] uses CLIP to generate pseudo-labels for single-image segmentation. STRICT [25] obtains pixel-level pseudo-labels from CLIP for unlabeled pixels and proposes a self-training strategy to capture latent information on unseen classes. LSeg [8] trains a CNN model to compute per-pixel image embeddings and use CLIP text embeddings as a classifier. [32] employs contrastive supervision to learn segmentation masks from text.

Concurrently, recent works [6; 33; 27; 20; 9] follow the "frozen CLIP" paradigm for zero-shot segmentation, they first generate a series of mask proposals and then utilize CLIP [28] or ALIGN [17] to classify them. ZSSeg and OVSeg [33; 20] train CLIP adapters to boost performance. FreeSeg[27] simultaneously uses semantic, instance, and panoptic labels and performs fusion training. OpenSeg[9] takes extra images with image-level supervision (*e.g.* captions) to scale up training data.

**Pre-trained model fine-tuning** is widely used for transferring pre-trained knowledge to downstream tasks, *e.g.* segmentation. However, this strategy may not work well for data-limited tasks like few-shot learning and zero-shot learning due to the daunting *overfitting* problem. To address this problem and transfer pre-trained knowledge to data-limited tasks, [43; 42; 12; 33; 20; 27] propose to learn text prompts or image prompts by using (a few) annotated images from target dataset. SVF [30] fine-tunes only a few parameters in the pre-trained image encoder to adapt pre-trained knowledge to few-shot segmentation. [38; 37] use contrastive learning to avoid catastrophic forgetting. Alternatively, many outstanding approaches in data-limited tasks [23; 35; 36; 6; 33] choose to freeze the parameters of pre-trained models to maintain the transferability.

Specific to the task of zero-shot/ open-vocabulary segmentation, mainstream approaches use frozen CLIP to avoid overfitting. Recently, MaskCLIP [41] conducts adequate experiments to fine-tune CLIP for open-vocabulary segmentation but has failed. While this attempt is meaningful and appreciated, it is believed that the failure is due to the large domain gap between pixel-level and image-level tasks. This motivates us further research fine-tuning CLIP to be mask-aware (region-level task).

## 3 Preliminary

**Problem Setting.** Zero-shot segmentation aims at training a segmentation model capable of segmenting novel objects using text descriptions. Given two category sets $C_{seen}$ and $C_{unseen}$ respectively, where $C_{seen}$ and $C_{unseen}$ are disjoint in terms of object categories ($C_{seen} \cap C_{unseen} = \emptyset$). The model is trained on $C_{seen}$ and directly tested on both $C_{seen}$ and $C_{unseen}$. Typically, $C_{seen}$ and $C_{unseen}$ are described with semantic words (*e.g.* sheep, grass).

**Revisiting the "frozen CLIP" paradigm.** The "frozen CLIP" approaches [6; 33; 27; 20] execute zero-shot segmentation in two steps: mask proposals generation and mask proposals classification. In the first step, these approaches train a Proposal Generator to generate $N$ class-agnostic mask proposals (denoting as $M$, $M \in \mathbb{R}^{N \times H \times W}$) and their corresponding classification scores (denoting as $A^p$, $A^p \in \mathbb{R}^{N \times |C_{seen}|}$). MaskFormer [5] and Mask2Former [4] are generally used as the Proposal Generator since the Hungarian matching [19] in the training process makes the mask proposals strongly generalizable. In the second step, $N$ suitable sub-images ($I_{sub}$) are obtained by *merging* $N$ mask proposals and the input image. $I_{sub}$ is then fed into the CLIP Image Encoder to obtain the image embedding ($E^I$). Meanwhile, text embedding ($E^T$) is generated by a CLIP Text Encoder. The classification score ($A^c$, $A^c \in \mathbb{R}^{N \times C}$) predicted by CLIP is calculated as:

$$A_i^c = \text{Softmax}(\frac{\exp(\frac{1}{\tau} s_c(E_i^T, E^I))}{\sum_{i=0}^{C} \exp(\frac{1}{\tau} s_c(E_i^T, E^I))}), i = [1, 2, ...C] \tag{1}$$

where $\tau$ is the temperature hyper-parameter. $s_c(E_i^T, E^I) = \frac{E_i^T \cdot E^I}{|E_i^T||E^I|}$ represents the cosine similarity between $E_i^T$ and $E^I$. $C$ is the number of classes, with $C = |C_{seen}|$ during training and $C = |C_{seen} \cup C_{unseen}|$ during inference. Noting that CLIP is frozen when training to avoid overfitting.

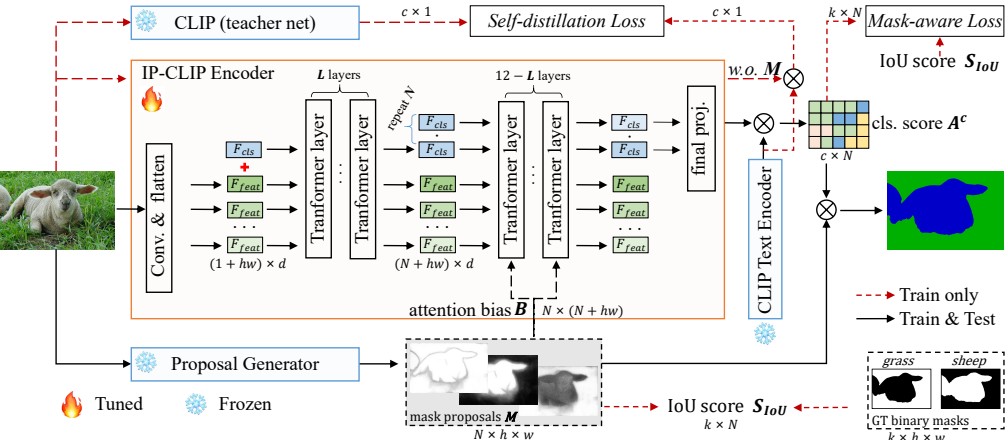

Figure 2: Overview of the Mask-Aware Fine-tuning (MAFT). In IP-CLIP Encoder, we modify the CLIP Image Encoder, and apply the mask proposals as attention bias in Multihead Attention from the $L^{th}$ layer. The final projection unit is an MLP module used for reshaping the channels of $F_{cls}$. *w.o.* $M$ denotes IP-CLIP Encoder processes image without utilizing mask proposals ($M$). *Mask-aware* Loss is designed to train CLIP to be mask-aware, while *Self-distillation* Loss is designed to maintain the transferability. Only the IP-CLIP Encoder is trained (orange part), the Proposal Generator and the CLIP Text Encoder are frozen (blue part).

To further enhance the reliability of $A^c$, the classification score of the Proposal Generator ($A^p$) is ensembled with $A^c$ since $A^p$ is more reliable on seen classes. This *ensemble* operation is wildly used in "frozen CLIP" approaches. The pipeline of "frozen CLIP", as well as the *merge* and *ensemble* operations, are described in detail in the Appendix.

Although "frozen CLIP" approaches have achieved promising results, it is clear that directly adopting an image-level pre-trained CLIP for proposal classification can be suboptimal. A frozen CLIP usually produces numerous false positives, and the *merge* operation may destroy the context information of an input image. In view of this, we rethink the paradigm of the frozen CLIP and explore a new solution for proposal classification.

## 4 Methodology

We introduce Mask-Aware Fine-tuning (MAFT), a method for learning mask-aware CLIP representations. Within MAFT, we first propose the Image-Proposal CLIP Encoder (IP-CLIP Encoder) to handle images with any number of mask proposals simultaneously (Sec. 4.1). Then, *mask-aware loss* and *self-distillation loss* are introduced to fine-tune the IP-CLIP Encoder and make it distinguishable for different mask proposals while maintaining transferability (Sec. 4.2). The complete diagram of the MAFT is shown in Fig. 2, we use the ViT-B/16 CLIP model for illustration.

### 4.1 Image-Proposal CLIP Encoder (IP-CLIP Encoder)

IP-CLIP Encoder aims to process arbitrary numbers of images and mask proposals simultaneously. We draw inspiration from MaskFormer [4; 5], which uses attention-masks in Multihead Attention and provides the flexibility for accepting any number of queries and features of different masked regions. Accordingly, we apply mask proposals as attention-masks in Multihead Attention and designate independent classification queries for each mask proposal.

In the IP-CLIP Encoder shown in Fig. 2, we denote the features propagate between Transformer layers as $F^i$, where $i = [1, 2...12]$. We can express $F^i$ as $F^i = [F^i_{cls}; F^i_{feat}], \in \mathbb{R}^{(1+hw) \times d}$, here 1 represents a class-embedding vector ($F^i_{cls}$), $hw$ represents the number of the flattened image features ($F^i_{feat}$). To obtain the classifications of all mask proposals simultaneously, we repeat $F^i_{cls}$ at layer $L$ $N$ times, where $N$ is the number of mask proposals, denoting the repeated class-embedding vectors as $F^{i*}_{cls}$. We can express the modified features ($F^{i*}$) as $F^{i*} = [F^{i*}_{cls}; F^i_{feat}], \in \mathbb{R}^{(N+hw) \times d}$.

**Propagation of $F^i$, where $i = [1, 2, ...L]$.** We consider that CLIP's classification significantly relies on context information. In the first $L$ Transformer layers, the propagation of $F^i$ is the same as in standard CLIP. Specifically, $F^i_{cls}$ utilizes cross-attention with all pixels within $F^i_{feat}$, effectively retaining the context information.

In the subsequent $12 - L$ Transformer layers, the propagation of $F^{i*}$ can be partitioned into two parts: the propagation of $F^{i*}_{cls}$ and the propagation of $F^i_{feat}$.

**Propagation of $F^{i*}_{cls}$.** We use $F^{i*}_{cls}[n]$ and $M[n]$ to represent the position $n$ in $F^{i*}_{cls}$ and $M$, where $n = [1, 2...N]$. It is expected $F^{i*}_{cls}[n]$ computes Multihead Attention for the positions where $M[n] = 1$ and itself. To achieve this, we construct an attention bias $B \in \mathbb{R}^{N \times (N+hw)}$ as follows:

$$B_{(i,j)} = \begin{cases} 0, \text{if } \hat{M}_{(i,j)} = 1 \\ -\infty, \text{if } \hat{M}_{(i,j)} = 0 \end{cases}, \quad \hat{M} = [\text{I}(N, N); \text{ Flat}(M)] \tag{2}$$

here $\text{I}(N, N)$ denotes $N^{th}$ order identity matrix, $\text{Flat}(\cdot)$ denotes the *flatten* operation. $\hat{M}$ is an intermediate variable for better representation. Therefore, a masked Multihead Attention is used for propagating $F^{i*}_{cls}$ :

$$F^{(i+1)*}_{cls} = \text{Softmax}(\frac{\text{Que}(F^{i*}_{cls})\text{Key}(F^{i*})^T}{\sqrt{d}} + B)\text{Val}(F^{i*}) \tag{3}$$

where $\text{Que}(\cdot)$, $\text{Key}(\cdot)$, and $\text{Val}(\cdot)$ denote linear projections, $d$ is the hidden dimension of $F^{i*}$. Notably, We omit the MLP Layer and Layer Normalizations in Transformer layers to simplify the representation in Eq. 3 and Eq. 4.

**Propagation of $F^i_{feat}$.** A standard Multihead Attention is used for propagating $F^i_{feat}$ :

$$F^{i+1}_{feat} = \text{Softmax}(\frac{\text{Que}(F^i_{feat})\text{Key}(F^i_{feat})^T}{\sqrt{d}})\text{Val}(F^i_{feat}) \tag{4}$$

Therefore, for any given mask proposal $M[n]$, the corresponding class-embedding $F^{i*}_{cls}[n]$ only performs Multihead Attention with $F^i_{feat}$ where $M[n] = 1$ and $F^{i*}_{cls}[n]$. The propagation of $F^i_{feat}$ remains undisturbed by attention-masks. Compared with the frozen CLIP, IP-CLIP Encoder leverages context information effectively and reduces computational costs.

## 4.2 Objective

IP-CLIP Encoder with CLIP pre-trained parameters remains challenging in distinguishing different mask proposals, *e.g.*, when the proposals contain more background regions than foreground objects, IP-CLIP may tend to classify them into the foreground categories. To overcome this limitation, we introduce *mask-aware loss* and *self-distillation loss* to fine-tune the IP-CLIP Encoder to be mask-aware without sacrificing transferability.

We conduct the *mask-aware* loss function ($\mathcal{L}_{ma}$) on $A^c$. The goal is to assign high scores to high-quality proposals and low scores to low-quality proposals in $A^c$. Concretely, we use the Intersection over Union (IoU) score obtained from ground-truth and align it with the $A^c$ to prompt CLIP to become mask-aware. Assuming there are $k$ classes in ground-truth, we can generate $k$ binary maps of ground-truth and calculate the IOU score ($S_{IoU}$) with $N$ mask proposals. We identify a discrepancy between the maximum values of $A^c$ and $S_{IoU}$. The maximum value of $A^c$ tends to approach 1, whereas the maximum value of $S_{IoU}$ ranges from 0.75 to 0.99. This inconsistency can hinder the alignment between these two metrics. Therefore, we introduced a min-max normalization technique for $S_{IoU}$ as follows:

$$S^{norm}_{IoU} = \frac{S_{IoU} - min(S_{IoU})}{max(S_{IoU}) - min(S_{IoU})}, S_{IoU} \in \mathbb{R}^{K \times N} \tag{5}$$

Meanwhile, we select $k$ pre-existing classes in $A^c$ ($A^c_{select}, A^c_{select} \in \mathbb{R}^{K \times N}$), and employ $SmoothL1$ Loss to align it with $S^{norm}_{IoU}$. Therefore, $\mathcal{L}_{ma}$ can be formulated as follows:

$$\mathcal{L}_{ma}(A^c_{select}, S^{norm}_{IoU}) = \text{SmoothL1}(A^c_{select}, S^{norm}_{IoU}) \tag{6}$$

Table 1: Comparison with state-of-the-art methods in zero-shot segmentation. $mIoU^s$ and $mIoU^u$ denote the mIoU(%) of seen classes and unseen classes.

| Method | COCO-Stuff | | | Pascal-VOC | | | ADE20K | | |
|--------|--------|--------|--------|--------|--------|--------|--------|--------|--------|
| | $mIoU^s$ | $mIoU^u$ | hIoU | $mIoU^s$ | $mIoU^u$ | hIoU | $mIoU^s$ | $mIoU^u$ | hIoU |
| SPNet[31] | 34.6 | 26.9 | 30.3 | 77.8 | 25.8 | 38.8 | - | - | - |
| ZS5[1] | 34.9 | 10.6 | 16.2 | 78.0 | 21.2 | 33.3 | - | - | - |
| CaGNet[10] | 35.6 | 13.4 | 19.5 | 78.6 | 30.3 | 43.7 | - | - | - |
| STRICT[25] | 35.3 | 30.3 | 32.6 | 82.7 | 35.6 | 73.3 | - | - | - |
| ZegFormer[6] | 36.7 | 36.2 | 36.4 | 90.1 | 70.6 | 79.2 | 17.4 | 5.1 | 7.9 |
| ZegFormer +MAFT | 36.4 $_{-0.3}$ | 40.1 $_{+3.9}$ | 38.1 $_{+1.7}$ | 91.5 $_{+1.4}$ | 80.7 $_{+10.1}$ | 85.7 $_{+6.5}$ | 16.6 $_{-0.8}$ | 7.0 $_{+1.9}$ | 9.8 $_{+1.9}$ |
| ZSSeg[33] | 40.4 | 36.5 | 38.3 | 86.6 | 59.7 | 69.4 | 18.0 | 4.5 | 7.2 |
| ZSSeg +MAFT | 40.6 $_{+0.2}$ | 40.1 $_{+3.6}$ | 40.3 $_{+2.0}$ | 88.4 $_{+1.8}$ | 66.2 $_{+6.5}$ | 75.7 $_{+6.3}$ | 18.9 $_{+0.9}$ | 6.7 $_{+2.2}$ | 9.9 $_{+2.7}$ |
| FreeSeg[27] | 42.4 | 42.2 | 42.3 | 91.9 | 78.6 | 84.7 | 22.3 | 4.4 | 7.3 |
| FreeSeg +MAFT | 43.3 $_{+0.9}$ | 50.4 $_{+8.2}$ | 46.5 $_{+4.2}$ | 91.4 $_{-0.5}$ | 81.8 $_{+3.2}$ | 86.3 $_{+1.6}$ | 21.4 $_{-0.9}$ | 8.7 $_{+4.3}$ | 12.4 $_{+5.1}$ |

Table 2: Results on representative methods [6; 33; 27] with/without MAFT. Here we remove the *ensemble* operation, and only maintain CLIP classifier results.

| Method | COCO-Stuff | | | Pascal-VOC | | | ADE20K | | |
|--------|--------|--------|--------|--------|--------|--------|--------|--------|--------|
| | $mIoU^s$ | $mIoU^u$ | hIoU | $mIoU^s$ | $mIoU^u$ | hIoU | $mIoU^s$ | $mIoU^u$ | hIoU |
| ZegFormer[6] | 18.5 | 23.0 | 20.5 | 81.4 | 76.8 | 79.0 | 5.1 | 2.6 | 3.5 |
| ZegFormer +MAFT | 35.1 $_{+16.6}$ | 31.6 $_{+7.6}$ | 33.3 $_{+12.7}$ | 87.6 $_{+6.2}$ | 79.9 $_{+3.1}$ | 83.5 $_{+4.5}$ | 15.8 $_{+10.8}$ | 7.0 $_{+4.4}$ | 9.8 $_{+6.3}$ |
| ZSSeg[33] | 20.6 | 27.4 | 23.6 | 82.0 | 71.2 | 76.2 | 5.9 | 2.8 | 3.9 |
| ZSSeg +MAFT | 36.1 $_{+15.5}$ | 35.9 $_{+8.3}$ | 36.0 $_{+12.4}$ | 87.1 $_{+5.1}$ | 76.1 $_{+4.9}$ | 81.2 $_{+5.0}$ | 17.2 $_{+11.3}$ | 7.2 $_{+4.4}$ | 10.2 $_{+6.3}$ |
| FreeSeg[27] | 22.3 | 29.3 | 25.3 | 87.4 | 74.7 | 80.5 | 6.5 | 2.8 | 3.9 |
| FreeSeg +MAFT | 40.1 $_{+17.8}$ | 49.7 $_{+20.4}$ | 44.4 $_{+19.1}$ | 90.4 $_{+3.0}$ | 84.7 $_{+10.0}$ | 87.5 $_{+7.0}$ | 21.3 $_{+14.8}$ | 8.7 $_{+5.9}$ | 12.2 $_{+8.3}$ |

$$\text{SmoothL1}(x, y) = \begin{cases} 0.5 \cdot (x-y)^2, & \text{if } |x-y| < 1 \\ |x-y| - 0.5, & \text{otherwise} \end{cases} \quad (7)$$

In addition to $\mathcal{L}_{ma}$, we also introduce a *self-distillation* loss $\mathcal{L}_{dis}$ to maintain CLIP's transferability and alleviate overfitting on $C_{seen}$. Within $\mathcal{L}_{dis}$, we use a frozen CLIP as the *teacher* net, the IP-CLIP as the *student* net for self-distillation. The predictions of the frozen CLIP and IP-CLIP are expected to be the same when no mask is included. Denoting the output of the frozen CLIP as $A_T$, and the output of the fine-tuned IP-CLIP without masks as $A_S$. We use $SmoothL1$ Loss to minimize the difference as follows:

$$\mathcal{L}_{dis}(A_S, A_T) = \text{SmoothL1}(A_S, A_T) \quad (8)$$

It is important to note that when processing an image through IP-CLIP without mask proposals, the resulting $A_S$ is a matrix with dimensions $\mathbb{R}^{C \times 1}$. Therefore, the final loss function can be formulated as: $\mathcal{L} = \mathcal{L}_{ma} + \lambda \mathcal{L}_{dis}$, where we set the constant $\lambda$ to 1 in our experiments. The mask-aware fine-tuning process is efficient as we only perform a few iterations (less than 1 epoch).

## 5 Experiments

### 5.1 Setting

**Dataset.** We first follow [1; 11; 26; 6; 33] to conduct experiments on three popular zero-shot segmentation benchmarks, Pascal-VOC, COCO-Stuff and ADE20K, to evaluate our method. Then, we evaluate MAFT on the *open-vocabulary* setting [20; 33], *i.e.*, training on COCO-Stuff and testing on ADE20K (A-847, A-150), Pascal-Context (PC-459, PC-59), and Pascal-VOC (PAS-20). More details of the dataset settings are provided in the Appendix.

**Evaluation Metrics.** To quantitatively evaluate the performance, we follow standard practice [1; 31; 10; 25; 6; 33; 27], adopt mean Intersection over Union (mIoU) to respectively evaluate the performance for seen classes ($IoU^s$) and unseen classes ($IoU^u$). We also employ the harmonic mean IoU (hIoU) among the seen and unseen classes to measure comprehensive performance.

**Methods.** Three representative methods are used to verify the generality of MAFT. We unify the three methods into the same framework, with all methods using ResNet101 as the backbone of Proposal Generator and ViT-B/16 CLIP model for a fair comparison.

Table 3: Comparison with state-of-the-art methods on the *open-vocabulary* setting. mIoU is used to evaluate the performance. * denotes additional training data is used.

| | A-847 | A-150 | PC-459 | PC-59 | PAS-20 |
|---|---|---|---|---|---|
| SPNet[31] | - | - | - | 24.3 | 18.3 |
| ZSSeg[1] | - | - | - | 19.4 | 38.3 |
| LSeg+[8] | 2.5 | 13.0 | 5.2 | 36.0 | 59.0 |
| OVSeg[20] | 7.1 | 24.8 | 11.0 | 53.3 | 92.6 |
| OpenSeg* [9] | 8.8 | 28.6 | 12.2 | 48.2 | 72.2 |
| FreeSeg[27] | 7.1 | 17.9 | 6.4 | 34.4 | 85.6 |
| FreeSeg +MAFT | 10.1 +3.0 | 29.1 +11.2 | 12.8 +6.4 | 53.5 +19.1 | 90.0 +4.4 |

- **ZegFormer** (CVPR 2022) [6] is an early adopter of the "frozen CLIP" paradigm. It uses MaskFormer as Proposal Generator and employs an *ensemble* operation to improve the confidence of the results.
- **ZSSeg** (ECCV 2022) [33] uses MaskFormer as Proposal Generator and introduces learnable prompts to improve classification accuracy, which significantly affects the subsequent methods. ZSSeg also adopts a self-training strategy, this strategy is excluded from all methods for a fair comparison.
- **FreeSeg** (CVPR 2023) [27] represents the state-of-the-art method, unifies semantic, instance, and panoptic segmentation tasks and uses annotations from all three tasks for fusion training. We retrain FreeSeg with only the semantic annotations to ensure fairness.

**Implementation details.** We employ ResNet101 as backbone of the Proposal Generator and ViT-B/16 CLIP model. The training process consists of two stages. For the **first** stage, we follow the official code of ZegFormer, ZSSeg and FreeSeg for model training. For the **second** stage, we fine-tune IP-CLIP Encoder with MAFT. We take the batch size of 16 and set CLIP input image size to $480 \times 480$. The optimizer is AdamW with a learning rate of 0.00001 and weight decay of 0.00001. The number of training iterations is set to 100 for Pascal-VOC, 1000 for COCO-Stuff and 5000 for ADE20K.

## 5.2 Comparisons with State-of-the-art Methods

In this section, three representative methods are used [6; 33; 27] to evaluate the effectiveness of MAFT. We compare three representative methods with MAFT and frozen CLIP. Additionally, we compare the results with previous state-of-the-art methods [31; 1; 10; 25].

**Comparisons in the *zero-shot* setting.** In Tab. 1, MAFT remarkably improves the performance. MAFT promotes the state-of-the-art performance by + 8.2% on COCO, + 3.2% on Pascal, and +4.3% on ADE20K in terms of mIoU for unseen classes. It is important to note that the results for seen classes are mainly based on $A^p$ rather than $A^c$ due to the *ensemble* operation in [6; 33; 27] (Details in Sec. 3). Therefore, the effect of MAFT on the seen classes is relatively insignificant.

**Comparisons without ensemble strategy.** To better showcase the performance gains from MAFT, we removed the *ensemble* operation in [6; 33; 27] and presented the results in Tab. 2. It can be seen that the performance of different methods is significantly improved after applying MAFT. In particular, the state-of-the-art method FreeSeg achieves hIoU improvements of 19.1%, 7.0%, and 8.3% on COCO, VOC2012 and ADE20K datasets.

**Comparisons in the *open-vocabulary* setting.** We further evaluated the transferability of MAFT in the *open-vocabulary* setting [20; 33], using FreeSeg as a baseline for comparison. Results are shown in Tab. 3. Compared with OVSeg [20] and OpenSeg [9], FreeSeg achieves suboptimal performance. However, the proposed MAFT enhances the performance of A-847, A-150, PC-459, PC-59 and PAS-20 by 3.0%,11.2%, 6.4%, 19.1% and 4.4%, and outperforms OpenSeg on all five datasets.

## 5.3 Ablation Study

We conduct ablation studies on various choices of designs of our MAFT to show their contribution to the final results in Tab. 4. FreeSeg is used as the baseline model and *ensemble* operation is removed.

**Component-wise ablations.** To understand the effect of each component in the MAFT, including the IP-CLIP Encoder and the fine-tuning strategy ($\mathcal{L}_{ma}$, $\mathcal{L}_{dis}$), we start with standard FreeSeg and

Table 4: **Ablations on COCO dataset.** GFLOPs in (a) is used to measure the computation of CLIP Image Encoder. The best results are highlighted with red, and the default settings are highlighted with gray background.

(a) Ablation on components of **MAFT**. $ft$ denotes the mask-aware fine-tining

| | mIoU$^s$ | mIoU$^u$ | hIoU | GFLOPs |
|---|---|---|---|---|
| FreeSeg | 22.3 | 29.3 | 25.3 | 1127.0 |
| + IP-CLIP | 29.4 $_{+7.1}$ | 36.2 $_{+6.9}$ | 32.4 $_{+7.1}$ | 53.4 |
| + ft ($\mathcal{L}_{ma}$) | 39.9 $_{+17.6}$ | 47.1 $_{+17.8}$ | 43.1 $_{+17.8}$ | 53.4 |
| + ft ($\mathcal{L}_{ma} + \mathcal{L}_{dis}$) | 40.1$_{+17.8}$ | 49.7$_{+20.4}$ | 44.4$_{+19.0}$ | 53.4 |

(b) Ablation on **mask-aware loss** $\mathcal{L}_{ma}$. $\mathcal{L}_{dis}$ is removed.

| | mIoU$^s$ | mIoU$^u$ | hIoU |
|---|---|---|---|
| $L_1$ | 38.6 $_{+16.3}$ | 45.8 $_{+16.5}$ | 41.8 $_{+16.5}$ |
| $L_2$ | 40.0 $_{+17.7}$ | 45.8 $_{+16.5}$ | 42.7 $_{+17.4}$ |
| $SmoothL_1$ | 39.9 $_{+17.6}$ | 47.1 $_{+17.8}$ | 43.1 $_{+17.8}$ |
| $KL$ | 40.9 $_{+18.6}$ | 41.8 $_{+12.5}$ | 41.3 $_{+16.0}$ |

(c) Ablation of the **training iterations**

| | mIoU$^s$ | mIoU$^u$ |
|---|---|---|
| 500 iters | 37.7 | 47.0 |
| 1k iters | 40.0 | 49.7 |
| 2k iters | 41.1 | 47.6 |
| 3k iters | 41.4 | 46.5 |
| 4k iters | 41.5 | 46.1 |
| 5k iters | 42.0 | 45.7 |

(d) Ablation of the **frozen units in CLIP**

| | mIoU$^s$ | mIoU$^u$ | hIoU |
|---|---|---|---|
| None | 40.6 | 44.7 | 42.5 |
| + $cls.$ | 40.7 | 44.7 | 42.7 |
| + $pos.$ | 40.6 | 44.9 | 42.8 |
| + $mlp$ | 40.3 | 48.7 | 44.1 |
| + $conv.$ | 40.0 | 49.7 | 44.3 |
| + $proj.$ | 40.2 | 49.1 | 44.2 |

(e) Ablation of the **start mask attention layer** $L$

| | mIoU$^s$ | mIoU$^u$ | hIoU |
|---|---|---|---|
| 0 | 39.3 | 46.4 | 42.6 |
| 2 | 39.2 | 46.4 | 42.5 |
| 4 | 39.5 | 46.6 | 42.6 |
| 6 | 40.0 | 47.8 | 43.6 |
| 8 | 40.0 | 49.7 | 44.3 |
| 10 | 39.9 | 45.7 | 42.6 |

progressively add each design. (Tab. 4a). FreeSeg uses frozen CLIP and yields inferior performance due to CLIP's mask-unaware property ($1^{st}$ row). Then, IP-CLIP Encoder obtains rich context information and greatly reduces the omputational costs, resulting in an improvement of 7.1% on seen classes and 6.9% on unseen classes. However, mask-aware is not accomplished at this point. Using only $\mathcal{L}_{ma}$ for fine-tuning CLIP produces decent performance (the $3^{rd}$ result). The introduction of $\mathcal{L}_{dis}$ (the $4^{th}$ result) maintains transferability while learning mask-aware representations, which further enhances the performance on unseen classes by 2.6%.

**Effect of different $\mathcal{L}_{ma}$.** *Mask-aware* Loss $\mathcal{L}_{ma}$ is an essential component of MAFT. In Tab. 4b, we investigate how different loss functions ($L1$, $L2$, $SmoothL1$ and $KL$ Loss) impact performance, here we remove $\mathcal{L}_{dis}$ for analysis. Results show $SmoothL1$ Loss boosts performance on $C_{unseen}$ to 47.1% (+17.8%), $KL$ Loss provides +12.5% improvement on $C_{seen}$, but only +11.8% on $C_{unseen}$, manifesting $KL$ Loss compromises the model of transferability comparing with $SmoothL1$ Loss.

**Training iterations.** Tab. 4c examines the impact of training iterations. Increasing the number of iterations leads to gradual improvement of IoU$^s$, but it also results in significant overfitting on unseen classes. Therefore, we choose to fine-tune 1k iterations to maximize the zero-shot ability.

**Frozen units in CLIP.** We also explore the impact of fine-tuning units within IP-CLIP Encoder. As illustrated in Fig. 2, IP-CLIP Encoder comprises convolution layers (dubbed as $conv.$), class embedding ($cls.$), Transformer layers, final projection ($proj.$) and positional embedding ($pos.$, not shown in Fig. 2). We start with fine-tuning the entire IP-CLIP Encoder, and then freezing each unit sequentially, as specified in Tab. 4d. We only freeze $MLP$ in the Transformer layers (dubbed as $mlp$). Compared with fine-tuning the entire IP-CLIP Encoder, the performance of mIoU$^u$ is improved by 5.0% when freezing $conv.$, $cls.$, $pos.$ and $mlp$.

**Start mask attention layer**. Tab. 4e presents the results of the start mask attention layer ($L$). We observe a significant improvement in the performance of unseen classes by +3.4% when the value of $L$ increases from 0 to 8. This could be attributed to the fact that starting masked Multihead Attention later enables $F_{cls}^{i*}$ to gain more context information. However, the performance significantly drops when $L = 10$ (from 49.7% to 45.7%), which may be due to the loss of mask-aware property.

## 5.4 Extending MAFT with SAM

We explore using the Segment Anything Model [18] (SAM) as the proposal generator. We evaluate the performance with SAM-H using an original CLIP (dubbed $SAM$) or a mask-aware fine-tuned CLIP (dubbed $SAM + MAFT$). In fact, SAM can be seamlessly integrated into our framework as the proposal generator. The results are shown in Tab. 5. Experiments are conducted under both *zero-shot* setting and *open-vocabulary* setting.

It can be observed that $SAM + MAFT$ obtains significant improvement over SAM under both settings. Besides, $SAM + MAFT$ also surpasses FreeSeg + MAFT on all benchmarks. Particularly, in the zero-shot setting (Pascal-VOC), $SAM + MAFT$ outperforms FreeSeg + MAFT by 6.8% in

Table 5: Comparison with SAM. We use SAM-H as the proposal generator.

(a) Results in zero-shot segmentation.

|  | Pascal-VOC | | | | COCO-Stuff | | | |
|  | $mIoU^s$ | $mIoU^u$ | hIoU | mIoU | $mIoU^s$ | $mIoU^u$ | hIoU | mIoU |
|---|---|---|---|---|---|---|---|---|
| SAM | 85.1 | 86.7 | 85.9 | 85.5 | 43.1 | 43.3 | 43.2 | 42.1 |
| SAM + MAFT | 91.0 $_{+5.9}$ | 88.6 $_{+1.9}$ | 89.8 $_{+3.9}$ | 90.4 $_{+4.9}$ | 43.4 $_{+0.3}$ | 51.5 $_{+8.2}$ | 47.1 $_{+3.9}$ | 44.1 $_{+2.0}$ |

(b) Results in open-vocabulary segmentation.

|  | A-847 | A-150 | PC-459 | PC-59 | PAS-20 |
|---|---|---|---|---|---|
| SAM | 9.0 | 21.3 | 7.8 | 33.7 | 87.5 |
| SAM + MAFT | 12.7 $_{+3.7}$ | 33.0 $_{+11.7}$ | 16.2 $_{+8.4}$ | 59.0 $_{+25.3}$ | 92.7 $_{+5.2}$ |

terms of $mIoU^u$. This enhancement can be attributed to the stronger generalization capabilities of SAM for unseen classes.

## 5.5 Extending MAFT with more Vision-Language Models

Table 6: Comparison with more Vision-Language Models.

|  | backbone | A-847 | A-150 | PC-459 | PC-59 | PAS-20 |
|---|---|---|---|---|---|---|
| OVSeg [20] |  | 9.0 | 29.6 | 12.4 | 55.7 | 94.5 |
| FreeSeg [27] | ViT-L | 8.5 | 21.0 | 7.6 | 33.8 | 86.4 |
| FreeSeg + MAFT |  | 12.1 $_{+3.6}$ | 32.0 $_{+11.0}$ | 15.7 $_{+8.1}$ | 58.5 $_{+24.7}$ | 92.1 $_{+5.7}$ |
| FreeSeg [27] | Res50 | 5.3 | 15.5 | 5.4 | 28.2 | 87.1 |
| FreeSeg + MAFT |  | 8.4 $_{+3.1}$ | 27.0 $_{+11.5}$ | 9.9 $_{+4.5}$ | 50.8 $_{+22.6}$ | 89.0 $_{+1.9}$ |

In order to demonstrate the efficacy and robustness of MAFT, we conduct experiments using stronger (CLIP-ViT-L) and ResNet-based (CLIP-Res50) Vision-Language Models. The open-vocabulary results are shown in Tab. 6, we also include the results of OVSeg with CLIP-ViT-L for comparison.

**CLIP-ViT-L.** According to Tab. 6, FreeSeg with a standard CLIP-ViT-L model (dubbed FreeSeg) still can not achieve satisfactory results. However, by integrating our MAFT (dubbed FreeSeg + MAFT), the segmentation results are remarkably enhanced, thus establishing new state-of-the-art benchmarks.

**CLIP-Res50.** Our MAFT can easily adapted into ResNet-based models. Specifically, we modified the AttentionPool2d unit within CLIP-R50 Image Encoder. The mask proposals are introduced as attention bias ($B$) in Multihead Attention, with $F_{cls}$ being repeated N times. Notably in CLIP-R50, $F_{cls}$ is obtained via GlobalAveragePooling performing on $F_{feat}$. The results are presented in Tab. 6. The performance on all 5 datasets is improved by a large margin. FreeSeg + MAFT with CLIP-R50 achieves competitive results with some CLIP-ViT-B-based methods according to Tab. 3.

## 5.6 Qualitative Study

**Visualizations of typical proposals.** Fig. 3 shows frozen CLIP and mask-aware CLIP classifications of typical proposals, including high-quality proposals of foreground ($p_1$, $p_4$), high-quality proposals of background ($p_3$, $p_6$), a proposal with background noise ($p_2$), and a proposal containing part of the foreground ($p_5$). The proposal regions are highlighted in green or yellow.
Several observations can be obtained: (1) The frozen CLIP provides good predictions for $p_1$ and $p_4$. (2) The frozen CLIP assigns $p_2$ as $cat$ and $p_5$ as $horse$, with scores even higher than $p_1$, $p_4$, indicating the frozen CLIP cannot distinguish proposals containing information on the same objects. (3) The frozen CLIP fails to give correct predictions for $p_3$ and $p_6$, which may be due to the lack of context information. (4) Our mask-aware CLIP gives good predictions for high-quality proposals ($p_1$, $p_3$, $p_4$, $p_6$) and provides suitable predictions for $p_2$ and $p_5$.

**Qualitative analysis.** We show some visual examples in Fig. 4. Some segmentation results of FreeSeg contain background noise (*e.g.* the $1^{st}$ & $2^{nd}$ row, $3^{rd}$ column) or contain only part of the objects ($3^{rd}$ row, $3^{rd}$ column). In ADE20K-847 dataset, too many classes may lead to the anticipated results (last row, $3^{rd}$ column) with the frozen CLIP. Using a mask-aware CLIP to learn mask-aware representations can significantly improve these segmentation results, as evident from the last column.

More visual samples are shown in the Appendix.

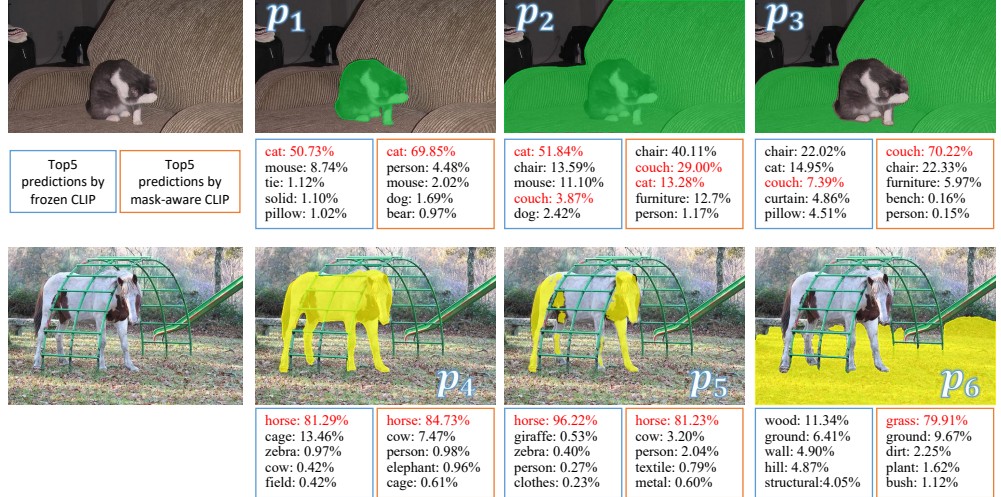

Figure 3: Visualizations of typical proposals & top 5 $A^c$ by frozen CLIP and mask-aware CLIP.

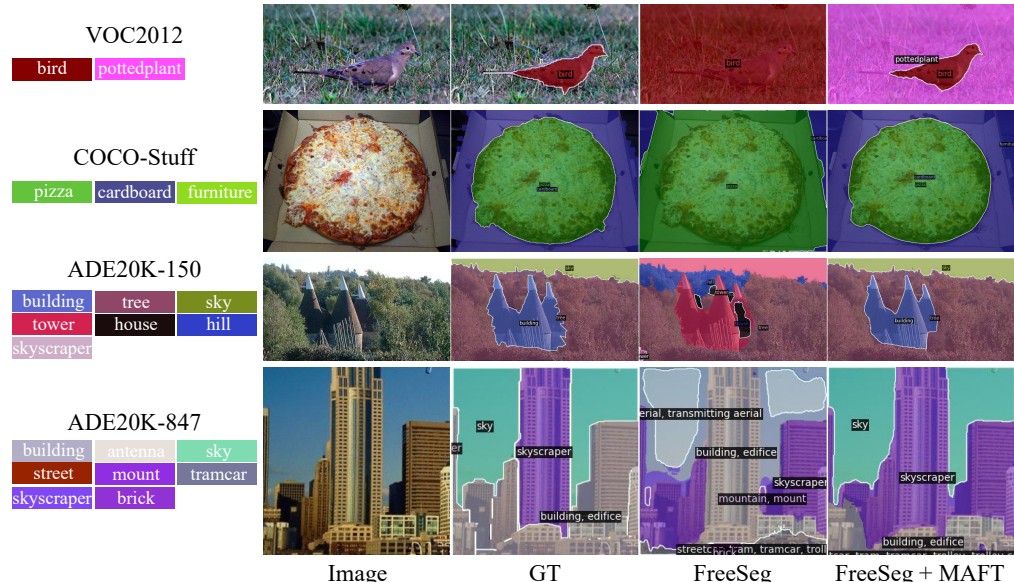

Figure 4: Qualitative results. The models are trained with COCO-Stuff and directly tested on VOC2012, COCO, and ADE20K.

## 6 Conclusion

In this paper, we rethink the "frozen CLIP" paradigm in zero-shot segmentation and propose Mask-Aware Fine-Tune (MAFT) for fine-tuning CLIP. Firstly, IP-CLIP Encoder is proposed to handle images with any number of mask proposals. Then, $\mathcal{L}_{ma}$ and $\mathcal{L}_{dis}$ are designed for fine-tuning CLIP to be mask-aware without sacrificing its transferability. MAFT is plug-and-play and can be applied to any "frozen CLIP" approach. Extensive experiments well demonstrate the performance of various zero-shot segmentation methods is improved by plugging MAFT.

**Limitations.** Our MAFT introduces a CLIP fine-tining framework to the research of zero-shot segmentation. However, the classification ability for novel classes is still limited by pre-trained vision-language models. How to further narrow this limitation is our future research focus.

**Acknowledgment.** This work was supported in part by the National Key R & D Program of China (No.2021ZD0112100).

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

# Appendix

Here we introduce technical details of the "frozen CLIP" approaches in Sec. A. The dataset settings are shown in Sec. B. Moreover, we provide additional experiments in Sec. C, and additional qualitative results in Sec. D.

## A  Technical details of the "frozen CLIP" approaches

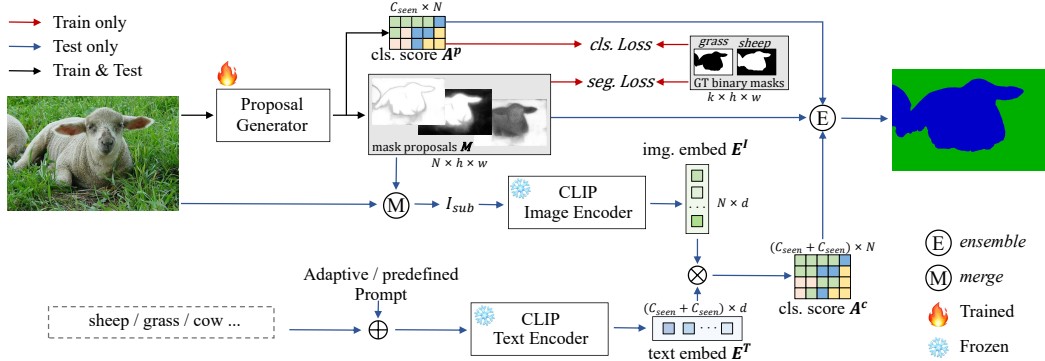

Figure 5: Overview of the "decoupling-paradigm".

Fig. 5 presents an overview of the "frozen CLIP" approach. **During training**, a standard MaskFormer or Mask2Former is used as Proposal Generator to generate $N$ mask proposals ($M$, $M \in \mathbb{R}^{N \times h \times w}$) and classification score ($A^p$, $A^p \in \mathbb{R}^{N \times |C_{seen}|}$). **During testing**, the input image is merged with $M$ to obtain $N$ sub-images ($I_{sub}$, $I_{sub} \in \mathbb{R}^{N \times \hat{h} \times \hat{w}}$). These sub-images are fed into a frozen CLIP to get the CLIP classification score ($A^c$, $A^c \in \mathbb{R}^{N \times |C_{seen} \cup C_{unseen}|}$). Here $C_{seen}$ and $C_{unseen}$ represent a set of seen classes and unseen classes. An *ensemble* operation is used to ensemble $A^p$ and $A^c$ for the final prediction. The *merge* and the *ensemble* operations will be introduced in detail in following:

**Merge operation.** To generate appropriate sub-images based on mask proposals, [6] presents three different *merge* operations: 1) mask, 2) crop, 3) mask & crop. Through experimentation, they demonstrate that the mask & crop option yields the best results. Fig. 6 provides an example of these operations. It's worth noting that all sub-images are resized to $\hat{h} \times \hat{w}$, here $\hat{h}$ and $\hat{w}$ typically take a value of 224, which is the default input size of CLIP Image Encoder. Although acceptable results can be obtained with the *merge* operation, it involves repeatedly feeding images into CLIP, which leads to significant computational redundancy.

**Ensemble operation.** Comparatively, $A^p$ provides higher confidence classification scores for the seen classes, and $A^c$ provides higher confidence classification scores for the unseen classes. Therefore, an ensemble of $A^p$ and $A^c$ achieves better results. The *ensemble* operation can be formulated as:

$$\hat{A}(c) = \begin{cases} A^p(c)^\lambda \cdot A^c(c)^{(1-\lambda)} \,, \ c \in C^{seen} \\ A^c(c)^\lambda \qquad\qquad\qquad , \ c \in C^{unseen} \end{cases} \qquad (9)$$

here a geometry mean of $A^p$ and $A^c$ is calculated (dubbed as $\hat{A}$), and the contribution of both classification scores is balanced by $\lambda$. As per literature [6; 33; 27], $\lambda$ usually takes values from 0.6 to 0.8. Therefore, the final output ($O$, $O \in \mathbb{R}^{|C_{seen} \cup C_{unseen}| \times h \times w}$) can be obtained by matrix multiplication: $O = \hat{A}^T \cdot M$. With the *ensemble* operation, the classification results of seen classes primarily depend on $A^p$, whereas the classification results of unseen classes mainly rely on $A^c$.

## B  Dataset

We follow [1; 11; 26; 6; 33] to conduct experiments on three benchmarks of the popular *zero-shot* setting, Pascal-VOC, COCO-Stuff and ADE20K, to evaluate the performance of MAFT. Additionally, we evaluate MAFT on the *open-vocabulary* setting [20; 33], *i.e.*, training on COCO-Stuff and testing on ADE20K, Pascal-Context, and Pascal-VOC.

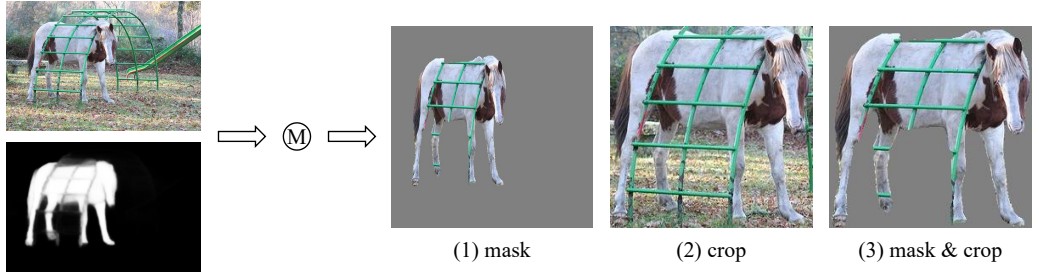

(1) mask      (2) crop      (3) mask & crop

Figure 6: Comparison among three *merge* operations.

- **COCO-Stuff**: COCO-Stuff is a large-scale semantic segmentation dataset that includes 171 classes. For the *zero-shot* setting [6; 33; 27], it is divided into 156 seen classes for training and 15 unseen classes for testing. For the *open-vocabulary* setting, all 171 classes are used for training.
- **Pascal-VOC**: There are 10582 images for training and 1,449 images for testing. For the *zero-shot* setting, Pascal-VOC is split into 15 seen classes and 5 unseen classes. For the *open-vocabulary* setting, all 20 classes are used for evaluation (dubbed as PAS-20).
- **ADE20K**: ADE20K contains 25k images for training and 2k images for validation. For the *zero-shot* setting, we follow [6] to choose 847 classes present in both training and validation sets, and split them into 572 seen and 275 unseen classes. For the *open-vocabulary* setting, we use two settings of ADE20K: 150 classes (dubbed as A-150) and 847 classes (dubbed as A-847).
- **Pascal-Context** is an extensive dataset of Pascal-VOC 2010. Two versions are used for *open-vocabulary* setting, one with 59 frequently used classes (dubbed as PC-59) and another with the whole 459 classes (dubbed as PC-459).

## C    Additional experiments

### C.1    Analysis of the Upper Bound of MAFT

Table 7: Upper Bound analysis.

|  | COCO-Stuff | | | |
| --- | --- | --- | --- | --- |
|  | $\text{mIoU}^s$ | $\text{mIoU}^u$ | hIoU | mIoU |
| MAFT | 43.3 | 50.4 | 46.5 | 43.9 |
| Upper Bound | 77.2 | 82.1 | 79.6 | 77.6 |

Considering the *mask-aware* loss may be limited if the quality of proposals is too bad, we conducted an evaluation of the Upper Bound while using Mask2Former as the proposal generator. The results are presented in Tab. 7. Specifically, we replace $A^c$ by $S_{IoU}$ (IoU score between binary ground-truth masks and proposals) during inference, and multiply proposals with $S_{IoU}$ to obtain the segmentation result. This result can be regarded as the Upper Bound for the given proposals. Notably, the Upper Bound achieves satisfactory results (77.6 % mIoU), indicating Mask2Former is capable of providing high-quality proposals in most cases. Additionally, there is still a large gap between the current performance and the Upper Bound ($\approx$ 30% mIoU), which suggests that our MAFT has enormous potential for improvement, whereas we have achieved state-of-the-art performance.

### C.2    Analysis of the Self-Training (ST) strategy

Several previous approaches [25; 33; 41] adopt the Self-Training (ST) strategy to enhance performance. we conduct experiments to investigate the application of ST into our method. Specifically, we use the existing FreeSeg + MAFT model to generate pseudo-labels for unseen classes on the training data, and then re-train FreeSeg with the pseudo-labels. Results are shown in Tab. 8.

Table 8: Analysis of the self-training (ST) strategy.

| | Pascal-VOC | | | | COCO-Stuff | | | |
|---|---|---|---|---|---|---|---|---|
| | $\mathbf{mIoU}^s$ | $\mathbf{mIoU}^u$ | **hIoU** | **mIoU** | $\mathbf{mIoU}^s$ | $\mathbf{mIoU}^u$ | **hIoU** | **mIoU** |
| MAFT | 91.4 | 81.8 | 86.3 | 89.0 | 43.3 | 50.4 | 46.5 | 43.9 |
| MAFT + ST | $90.0_{-1.4}$ | $86.3_{+4.5}$ | $88.1_{+1.8}$ | $89.1_{+0.1}$ | $44.1_{+0.8}$ | $55.2_{+4.8}$ | $49.0_{+2.5}$ | $45.0_{+1.1}$ |

The improvement of ST on the unseen category is significant (Pascal: 81.8 % → 86.3%, COCO: 50.4% → 55.2%) in terms of $\mathrm{mIoU}^u$. However, it's essential to highlight the applicability of ST is limited by a crucial requirement: **unseen classes need to be obtained during training.** This requirement poses significant limitations on generalizing ST to various scenarios, *e.g.*, open-vocabulary settings, since images of unseen classes may not be obtained during training.

## D  Visualization

We provide more qualitative results, including typical proposals and top-5 $A^c$ (Fig. 7), as well as examples of models trained on COCO-Stuff and text on A-847 (Fig. 8), A-150 (Fig. 9), PC-459 (Fig. 10), PC-59 (Fig. 11), Pascal-VOC (Fig. 12), and COCO-Stuff(Fig. 13).

**Typical Proposals and Top-5 $A^c$.** Fig. 7 shows frozen CLIP and mask-aware CLIP classifications of typical proposals. In the $2^{nd}$ column, we provide high-quality proposals of *thing* classes. Both the frozen CLIP and mask-aware CLIP provide high classification scores for the correct classes. In the $3^{rd}$ column, we provide proposals that only contain part of the objects (row 1-3), and proposals containing more than 1 class (row 4). The mask-aware CLIP provides more proper results compared to the frozen CLIP. In the $4^{th}$ column, we provide some high-quality background proposals. The frozen CLIP typically gives incorrect predictions, but the mask-aware CLIP assigns high scores for the correct classes.

**Qualitative Analysis.** Fig. 8,9,10,11,12,13 show segmentation results on Pascal-VOC, COCO-Stuff, ADE20K. In Pascal-VOC dataset (Fig. 12), which only contains 20 *thing* classes, the FreeSeg + MAFT model tends to assign background regions to the similar *thing* classes, *e.g.*, "train" in row 1, "pottedplant" in row3-4. "boat" in row 8. In A-847, A-150, PC-459, PC-59 and COCO-Stuff datasets, both seen classes and unseen classes exist in the input images, the FreeSeg + MAFT model generates better segmentation results compared to FreeSeg.

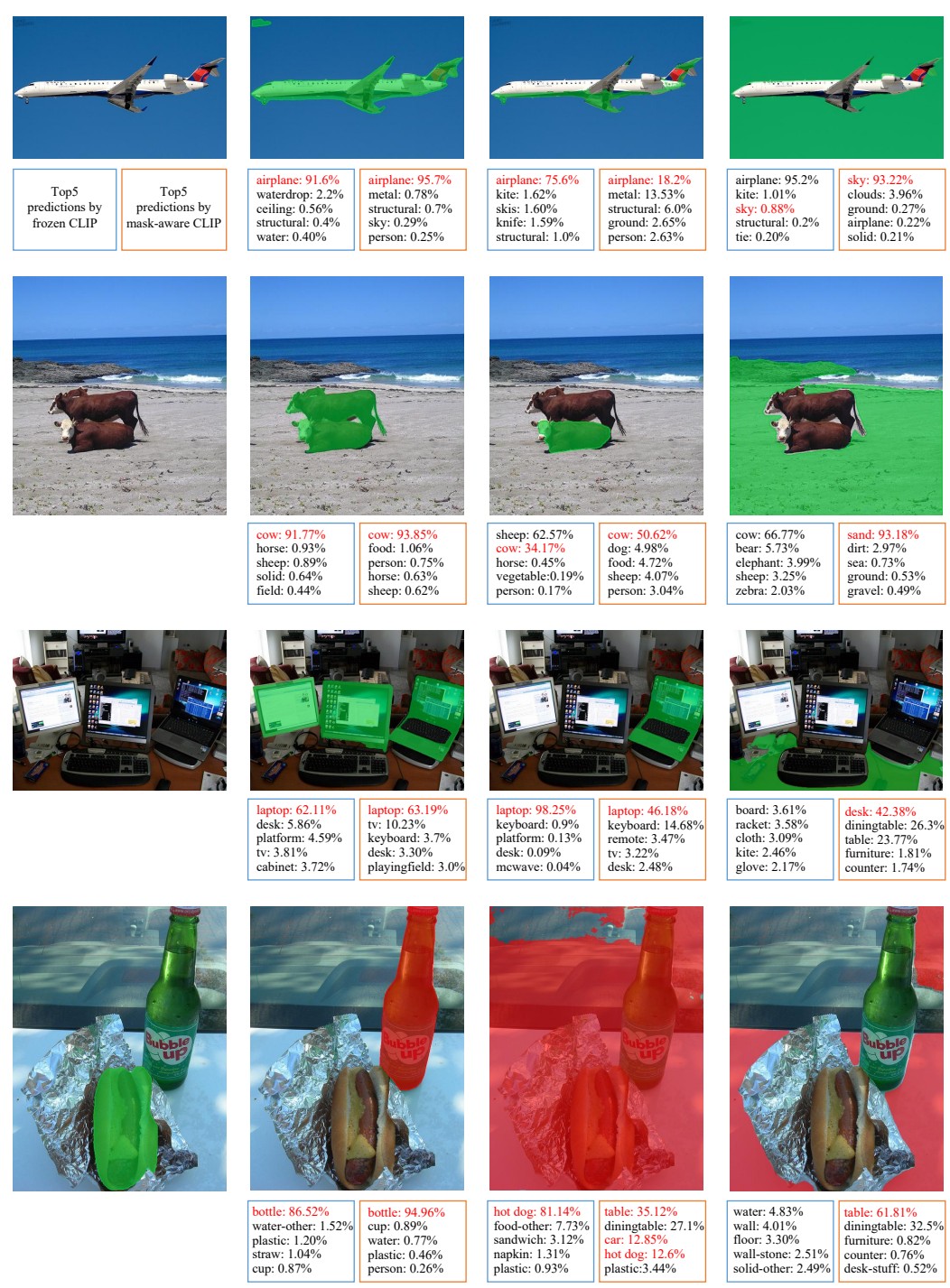

Figure 7: Visualizations of typical proposals & top 5 A$^c$ by frozen CLIP and mask-aware CLIP. The correct classes are highlighted in red.

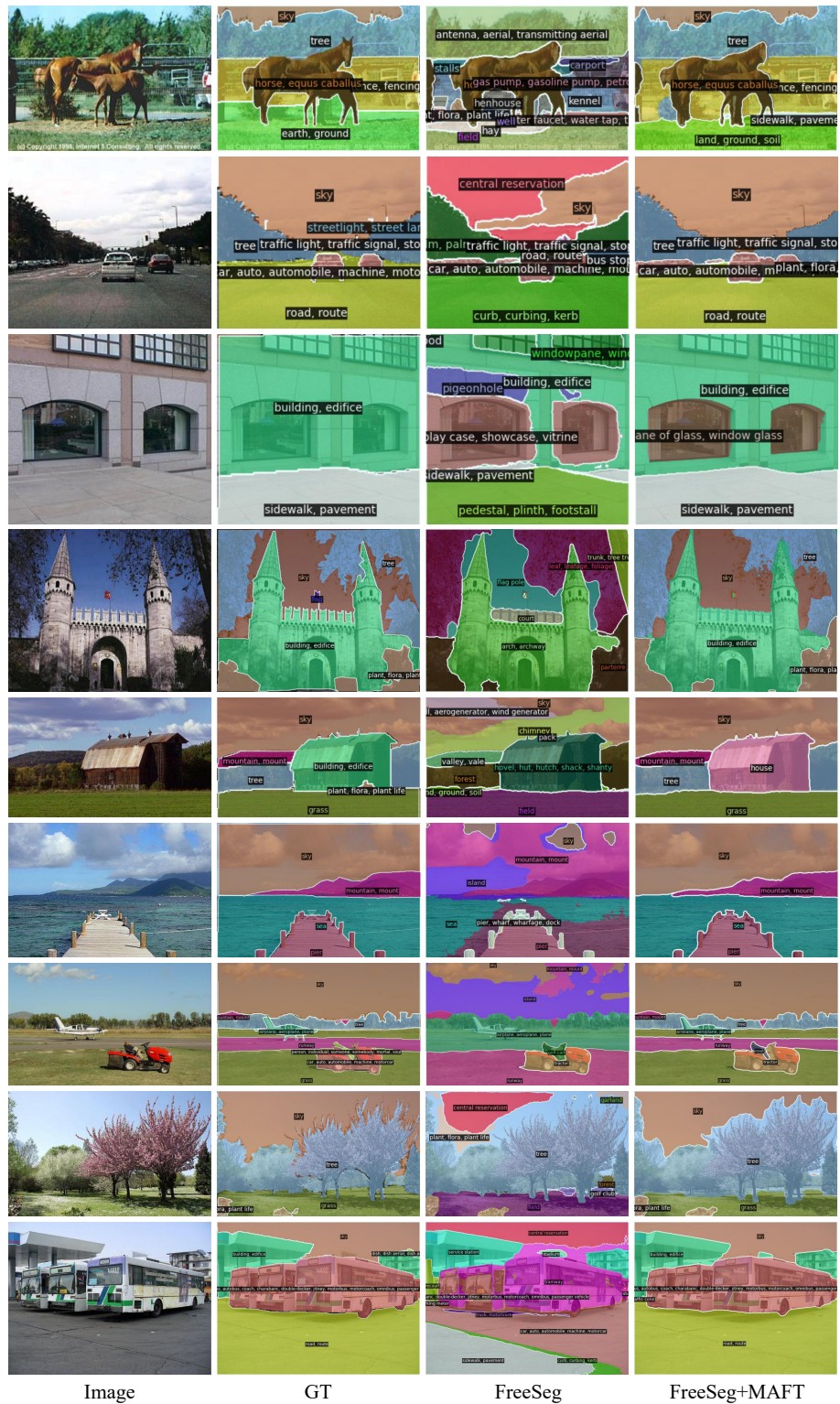

Figure 8: Qualitative results on A-847, using 847 class names in ADE20K to generate text embeddings.

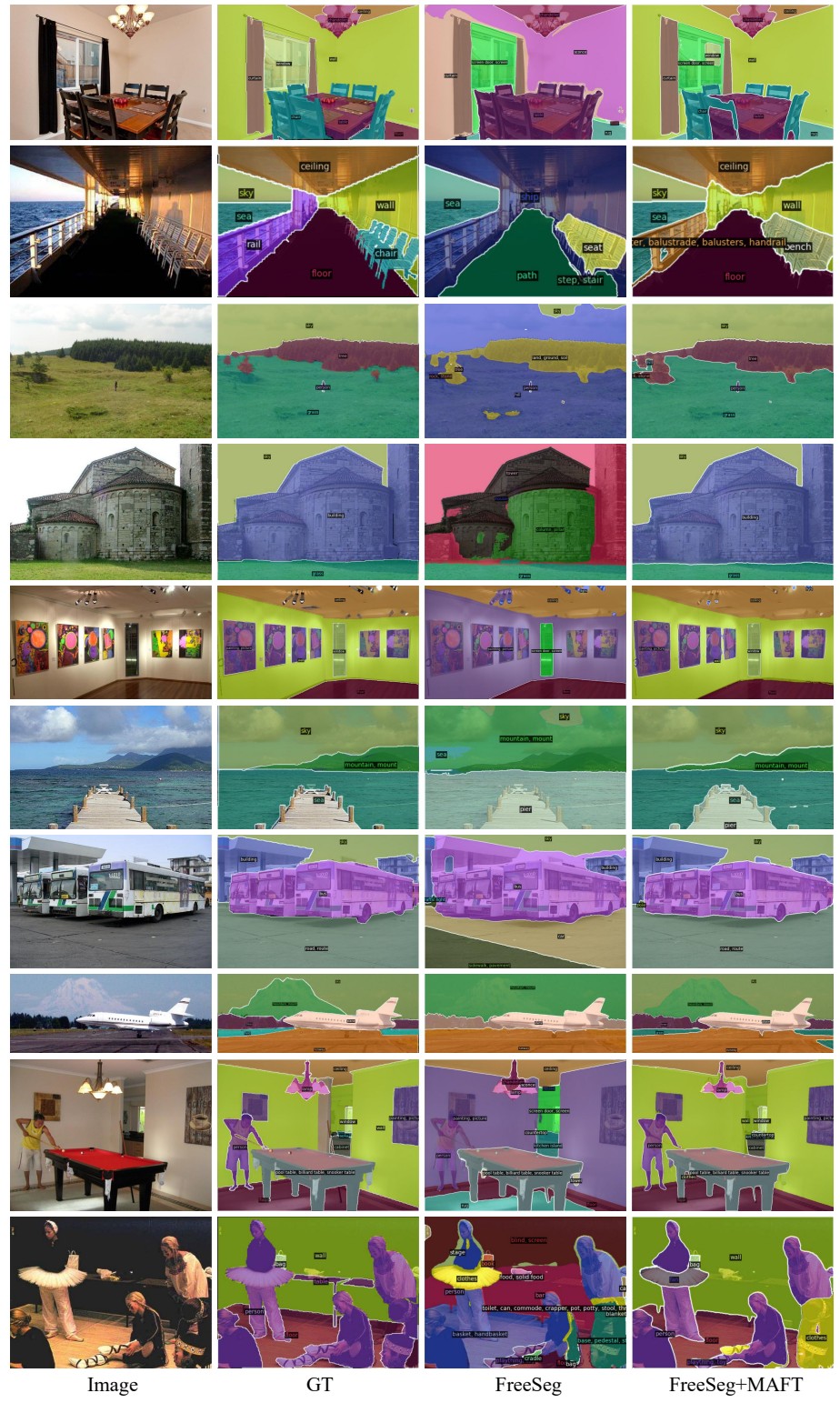

Figure 9: Qualitative results on A-150, using 150 class names in ADE20K to generate text embeddings.

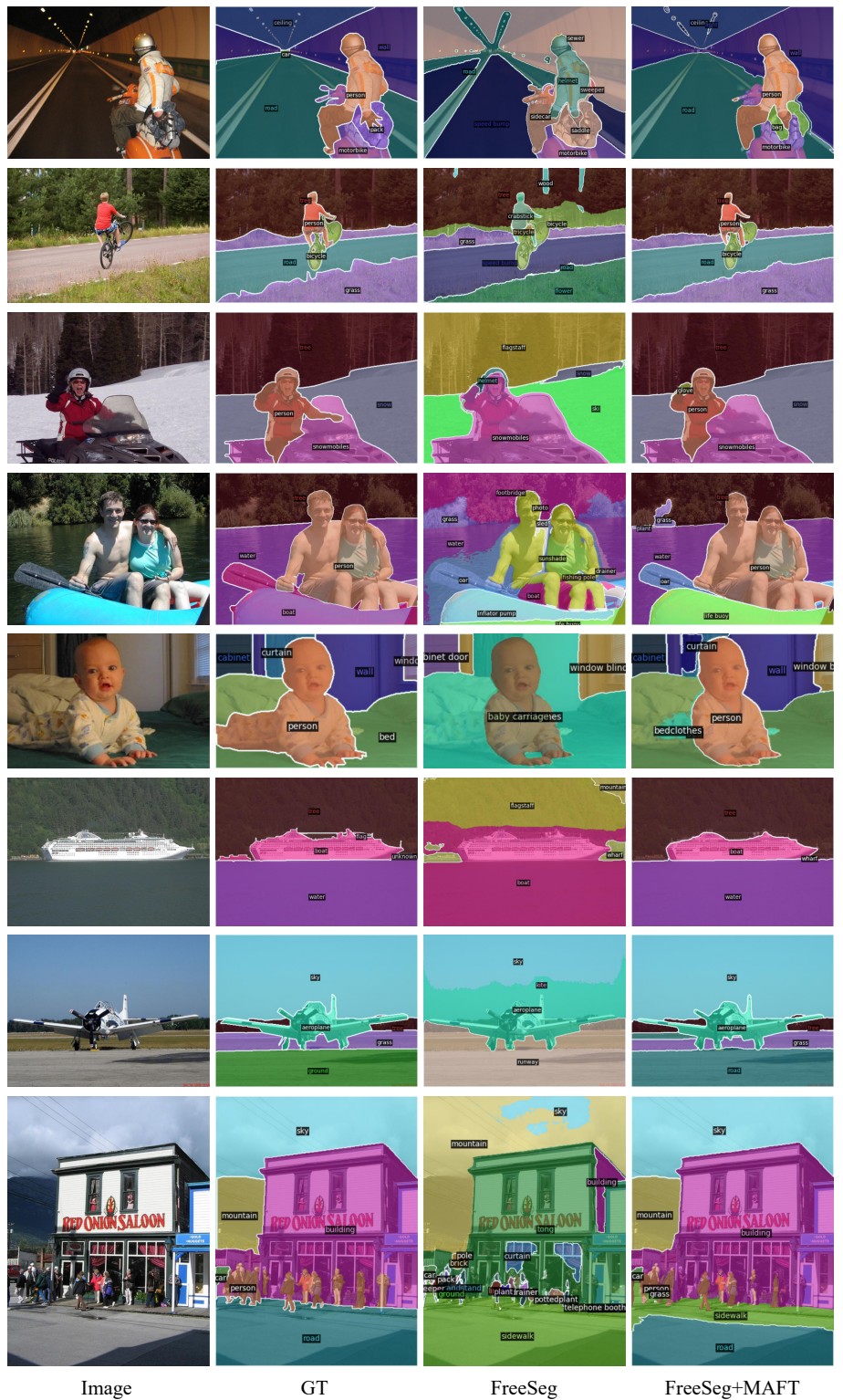

Image       GT       FreeSeg       FreeSeg+MAFT

Figure 10: Qualitative results on PC-459, using 459 class names in Pascal-Context to generate text embeddings.

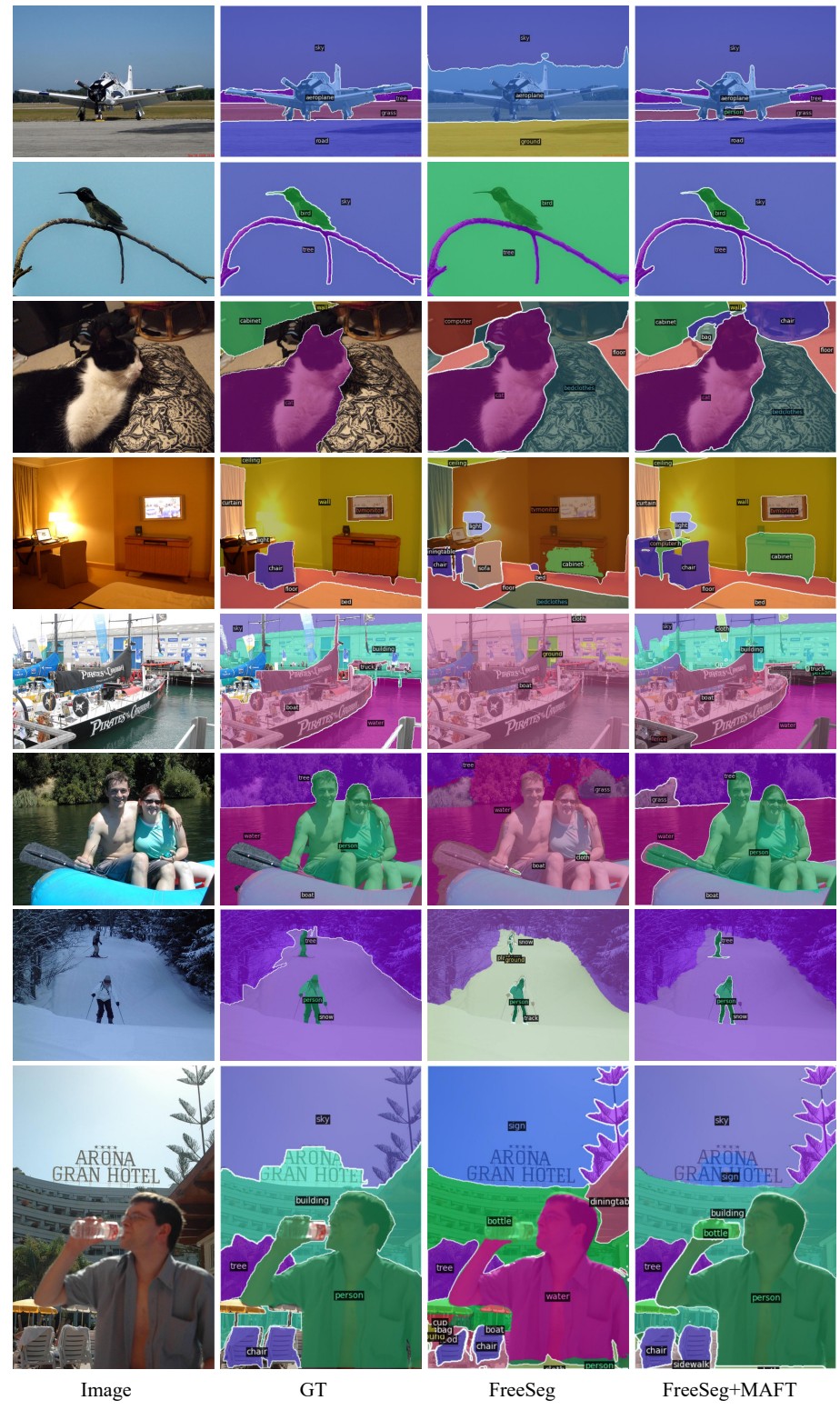

| Image | GT | FreeSeg | FreeSeg+MAFT |

Figure 11: Qualitative results on PC-59, using 59 class names in Pascal-Context to generate text embeddings.

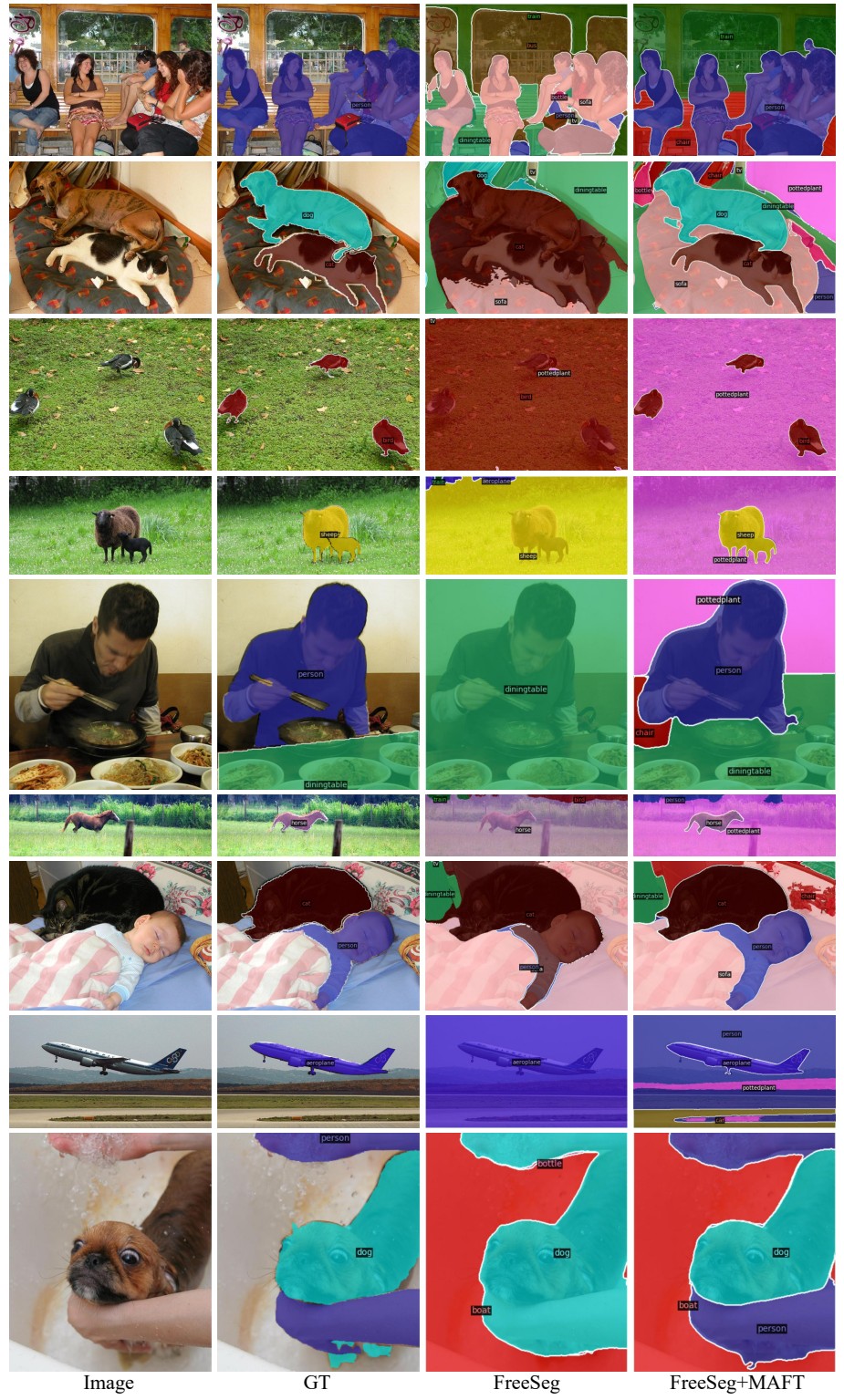

| Image | GT | FreeSeg | FreeSeg+MAFT |

Figure 12: Qualitative results on Pascal-VOC, using 20 class names in Pascal-VOC to generate text embeddings.

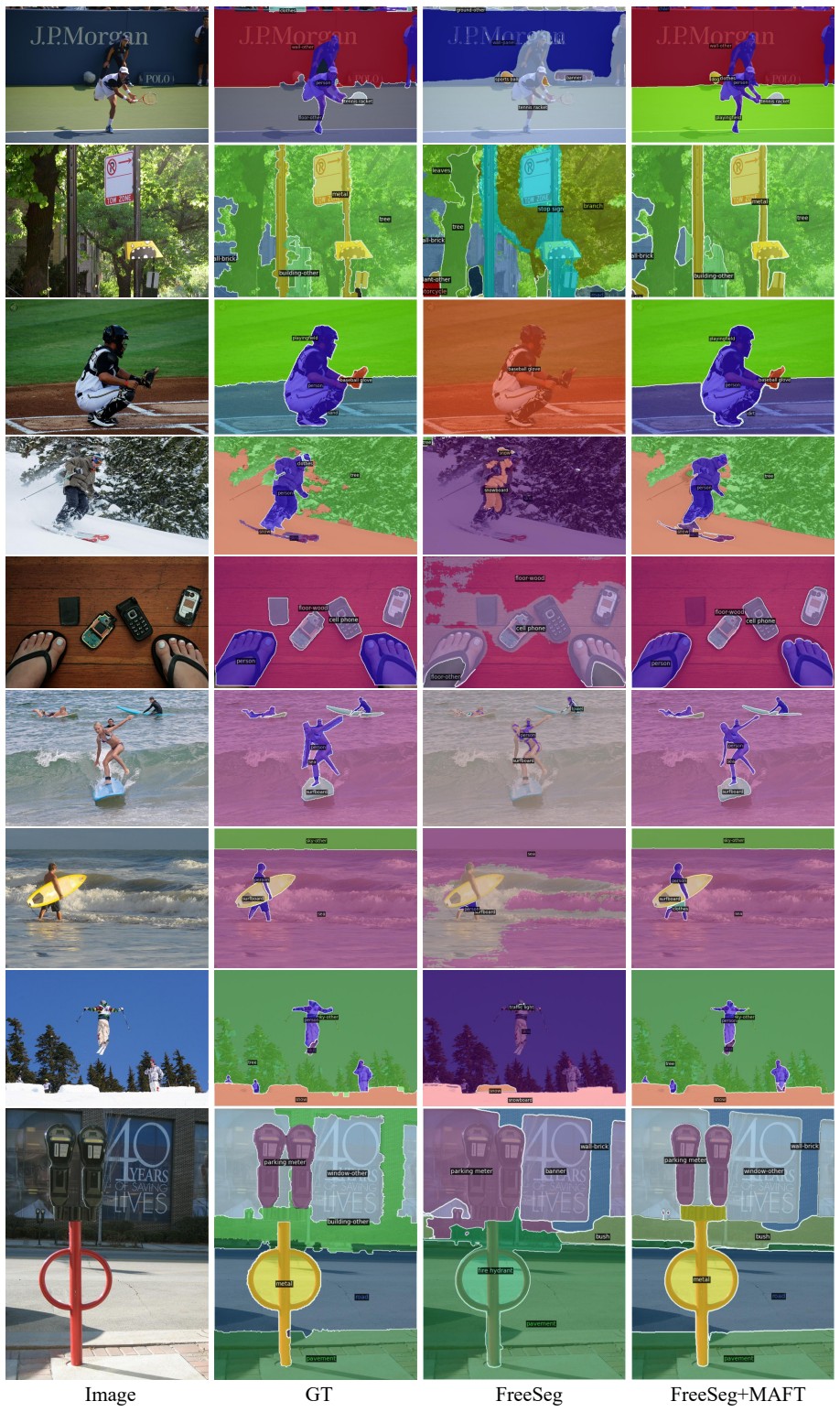

Figure 13: Qualitative results on COCO, using 171 class names in COCO-Stuff to generate text embeddings.

