# OpenReview forum: "Learning Mask-aware CLIP Representations for Zero-Shot Segmentation"
_NeurIPS.cc/2023/Conference — NeurIPS 2023 poster_

### Official Review · Reviewer_XVP7 · 2023-07-04

**Soundness:** 3 good
**Presentation:** 3 good
**Contribution:** 3 good
**Rating:** 7
**Confidence:** 5

**Summary:**

This paper proposes a new topic and method for training a mask-aware CLIP, which could serve as a core component for open-vocabulary segmentation. The designed structure could be used as a flexible plug-in but brings significant improvements for existing methods on various benchmarks.

**Strengths:**

1. This topic is promising.  Previous methods decouple open-vocabulary segmentation into class-agnostic segmentation and CLIP-guided recognition. However,  most of them fail to use CLIP effectively,  I think training a mask-aware CLIP is an ideal way to deal with this problem.

2. The model design is reasonable, using a mask2former-style network and tasks the masks to perform masked attention sounds reasonable.

3. The experiment results are great with significant improvement. It would be an ideal solution for various open-vocabulary segmentation tasks incorporating strong class-agnostic segmentation models like SAM.

4. The paper is clearly presented.

**Weaknesses:**

The experiment setting is unsatisfactory, which only tackles zero-shot semantic segmentation.
As this topic and idea are good,  I expect the authors to extend the method into open-vocabulary panoptic settings, and use some large datasets for training. Currently, the datasets used are small. it is hard to distill universal knowledge from CLIP.


**Questions:**

See weakness

**Limitations:**

yes

---

> ### Author Rebuttal · Authors · 2023-08-09
>
> # Response to Reviewer XVP7
>
> Thank you so much for acknowledging the strength of our method. We have carefully considered your constructive and insightful comments and here are the answers to your concerns.
>
> **Q1. Extending the method into open-vocabulary panoptic settings**
>
> First of all, we fully agree with you that MAFT could be applied to more expansive tasks, *e.g.*, open-vocabulary segmentation, rather than tackling zero-shot semantic segmentation only. Therefore, we have expanded our experimentation to open-vocabulary semantic segmentation, termed the "cross-dataset" setting within the paper. The results are shown in Tab. 3. The results substantiate our method's efficacy, as it outperforms existing approaches, thereby establishing a new state-of-the-art benchmark of open-vocabulary semantic segmentation.
>
> Furthermore, we start an initial exploration into open-vocabulary panoptic segmentation. We first train a FreeSeg model with COCO-Panoptic labels as the baseline. Then, we use the proposed IP-CLIP that has been fine-tuned solely with semantic labels for proposal classification. We experiment with both CLIP-ViT-B and CLIP-ViT-L, results are presented as follows:
>
>
> Open-vocabulary panoptic (PQ), instance (mAP) and semantic (mIoU) segmentation results with CLIP-ViT-B:
> ||PQ| mAP | mIoU
> | -------- | -------- | -------- |--- |
> | FreeSeg   | 10.5    |4.7 |18.2
> | FreeSeg + MAFT  | 17.1 (+6.6)   | 6.6 (+1.9)  | 27.6 (+9.4)
>
> Open-vocabulary panoptic (PQ), instance (mAP) and semantic (mIoU) segmentation results with CLIP-ViT-L:
> ||PQ| mAP | mIoU
> | -------- | -------- | -------- |--- |
> | FreeSeg   | 11.9    |7.1 |20.3
> | FreeSeg + MAFT  | 18.9 (+7.0)   | 9.2 (+2.1)  | 30.9 (+10.1)
>
> The results show the effectiveness of MAFT across different CLIP models in the open-vocabulary panoptic setting. For a comprehensive analysis of CLIP-ViT-L, please refer to the **Response to Reviewer q7Qn-Q1**. Notably, only semantic-level labels are used during the mask-aware fine-tuning process. It is possible to adapt *mask-aware loss* to panoptic-level labels, and would be an interesting future work.
>
> **Q2. Using some large datasets for training**
>
> In the standard open-vocabulary segmentation scenario, COCO serves for model training, while additional segmentation datasets, *e.g.*, ADE20K, PASCAL, and Cityscapes, are employed for testing
>
> Intuitively, utilizing larger datasets for training can lead to better performance. With this in mind, we are actively exploring further enriching our training pipeline by including datasets such as LVIS and Open Images. These datasets offer a wealth of diverse visual data, thus potentially enhancing the model's generalization capability and transferability.
>
>
> **Q3. Incorporating strong class-agnostic segmentation models like SAM**
>
> Thanks for your kind suggestions, we use SAM to instead of Mask2Former for proposal generation, resulting in notable performance enhancements for both open-vocabulary semantic segmentation and zero-shot segmentation. Please see **General Response-Q1** for details.

---

### Official Review · Reviewer_uMA5 · 2023-07-06

**Soundness:** 3 good
**Presentation:** 3 good
**Contribution:** 2 fair
**Rating:** 4
**Confidence:** 5

**Summary:**

This paper mainly discusses how to use pre-trained CLIP to solve zero-shot segmentation task, and proposes a new method called Mask-aware Fine-tuning (MAFT) to address the issue of significant false positives in CLIP's classification of mask proposals. Specifically, the paper introduces an Image-Proposals CLIP Encoder (IP-CLIP Encoder) to handle any number of images and mask proposals simultaneously, and designs mask-aware loss and self-distillation loss to fine-tune the IP-CLIP Encoder, ensuring that CLIP responds to different mask proposals without sacrificing its transferability.

**Strengths:**

1. This paper introduces an Image-Proposals CLIP Encoder (IP-CLIP Encoder) that is sensitive to different mask proposals.

2. This paper includes mask-aware loss and self-distillation loss to fine-tune the IP-CLIP Encoder without sacrificing its transferability.

3. The paper is well-written and easy to follow.

**Weaknesses:**

1. The ability to handle any number of mask proposals is not unique to this method and has already been a feature of previous methods such as ZegFormer.

2. The main effect of this method comes from the mask-aware loss, which utilizes mask proposals as prior knowledge to obtain more accurate prediction probabilities from the cls score map. Therefore, the effectiveness of this loss function is limited by the quality of the mask proposals, which limits the innovation of this paper.

3. In terms of experiments, it is necessary to conduct experiments on the updated methods such as "Scaling Open-Vocabulary Image Segmentation with Image-Level Labels"(ECCV2022) where the performance of it has already surpassed this method on VOC and COCO.

4. Why is Table 2's benchmark experiment conducted under the setting of using only CLIP classifier?

**Questions:**

Same as weakness.

**Limitations:**

The paper has a description of some limitations.

---

> ### Author Rebuttal · Authors · 2023-08-09
>
> # Response to Reviewer uMA5
>
> Thank you so much for acknowledging the strength of our method. We have carefully considered your constructive and insightful comments and here are the answers to your concerns.
>
> **Q1. The ability to handle any number of mask proposals is not unique.**
>
> There may be some misunderstandings about the contribution of our MAFT. Although there are a few approaches proposed to handle any number of mask proposals, the way our MAFT employs the mask proposals is significantly different from these "frozen CLIP" methods and is more effective and efficient.  We would like to re-emphasize the difference as follows.
>
> * The "frozen CLIP" approaches (*e.g.*, Zegformer, ZSSeg and FreeSeg) use a frozen CLIP for mask proposal classification. Specifically, they first use a $\mathrm{Merge}$ operation on the input image, creating $N$ sub-images ($I_{sub}$), where $N$ (defaulted to 100) is the number of mask proposals.This $\mathrm{Merge}$ operation is detailed at **Line 12** of the supplementary material. Then, $I_{sub}$ undergo classification with a frozen CLIP. This approach causes significant computational redundancy and the $\mathrm{Merge}$ operation loses image context information.
> * MAFT, on the other hand, innovates by modifying CLIP. It achieves simultaneous classification for all proposals with the image single pass through CLIP. Specifically, we modify CLIP by repeating $F_{cls}$ $N$ times and using $N$ mask proposals as attention bias in MultiHead Attention. This efficient design eliminates the need for the aforementioned $\mathrm{Merge}$ operation, reducing the computational cost and the image's context information is retained.
>
> To highlight the computational efficiency of MAFT, we provide GFLOPs analyses in Tab. 4(a). And we also evaluate the inference time (in terms of FPS) in the following table. All models are based on the CLIP-ViTB/16 and tested with one single RTX A6000 GPU for a fair comparison.
>
> |(FPS)|Zegformer|ZSSeg|FreeSeg
> |-|-|-|-
> |frozen CLIP|11.9|12.1|9.3
> |IP-CLIP|27.6|29.1| 25.5
>
> **Q2. Concerns about the quality of the mask proposals may limit the innovation**
>
> We agree that the *mask-aware loss* may be limited if the quality of proposals is too bad. However, we experimentally demonstrate that using Mask2Former as proposal generator can already provide high-quality proposals. In detail, we replace $A^{c}$ by $S_{IoU}$ (IoU score between binary groundtruth masks and proposals) during inference, and multiply proposals and $S_{IoU}$ to obtain the segmentation result. This result can be regarded as the Upper bound for the given proposals. Notably, the `Upper bound` achieves satisfactory results (77.6 mIoU), indicating Mask2Former can provide high-quality proposals in most cases. Additionally, there is still a large gap between current performance and the Upper bound (≈ 30% mIoU), which suggests that our MAFT has enormous potential for improvement whereas we have achieved state-of-the-art performance.
>
> Zero-shot setting (COCO-Stuff):
> ||mIoU$^{s}$|mIoU$^{u}$|hIoU|mIoU
> |-|-|-|-|-
> |FreeSeg + MAFT| 43.3|50.4|46.5|43.9
> |Upper bound|77.2|82.1|79.6|77.6
>
> Besides, we also explore using stronger proposal generators, *e.g.* SAM-H, as detailed in **General Response-Q1**. The results showcase that the MAFT approach achieves remarkable performance improvements when leveraging superior-quality proposals.
>
> We hope these experiments and analyses can demonstrate the efficacy and potential of our approach. Your concerns have guided us to delve deeper in our approach, and we are confident that our method offers a valuable contribution to the field.
>
> **Q3. Suboptimal performance comparing with OpenSeg**
>
> We have compared our method with "Scaling Open-Vocabulary Image Segmentation with Image-Level Labels" (dubbed OpenSeg*) in Tab. 3. Although OpenSeg* additionally uses Image-level labeled data for training, our method outperforms it on all five datasets (A-847, A-150, PC-459, PC-59, PAS-20).
>
> We also investigate new work and find that MaskCLIP+[1*] achieves better performance under zero-shot setting compared to ours. The performance gain in MaskCLIP+ comes mainly from the Self-Training (ST) strategy, which can be easily applied into our method. Specifically, we use the existing FreeSeg+MAFT model to generate pseudo-labels for unseen classes on the training data, and then re-train FreeSeg and MAFT with the pseudo-labels.
>
> |Pascal-VOC|mIoU$^{s}$|mIoU$^{u}$|hIoU|mIoU
> |-|-|-|-|-
> |FreeSeg + MAFT|91.4|81.8|86.3|89.0
> |FreeSeg + MAFT + ST|**90.0**(-1.4)|**86.3**(+4.5)|**88.1**(+1.8)|**89.1**(+0.1)
> |MaskCLIP+|88.8|86.1|87.4|88.1
>
> |COCO-Stuff|mIoU$^{s}$|mIoU$^{u}$|hIoU|mIoU
> |-|-|-|-|-
> |FreeSeg + MAFT|43.3|50.4|46.5| 43.9
> |FreeSeg + MAFT + ST|**44.1**(+0.8)|**55.2**(+4.8)|**49.0**(+2.5)|**45.0**+(1.1)
> |MaskCLIP+|38.1|54.7|45.0| 39.6
>
> The improvement of ST on the unseen category is significant (Pascal: 81.8 $\rightarrow$ 86.3, COCO: 50.4 $\rightarrow$ 55.2) in terms of mIoU$^{u}$. However, it's essential to highlight the applicability of ST is limited by a crucial requirement: **unseen classes need to be obtained during training**. This requirement poses significant limitations on generalizing ST to general scenarios, *e.g.* open-vocabulary settings and cross-dataset settings, since images of unseen classes may not be obtained when training. Therefore, we do not recommend the use of ST, as mentioned in **Line 211**.
>
> [1*] Extract Free Dense Labels from CLIP (ECCV22)
>
> **Q4. Reasons for using only CLIP classifier in Table 2**
>
> Previous works like Zegformer, ZSSeg and FreeSeg $\mathrm{ensemble}$ $A^{c}$ (CLIP classification score) and $A^{p}$ (proposals generator classification score). The supplementary materials introduce the $\mathrm{ensemble}$ operation in detail. However, our method only optimizes $A^{c}$ through the mask-aware fine-tuning process. So the setting of using only CLIP classifier in Tab. 2 provides a more direct and meaningful evaluation of the advancements achieved by MAFT.

---

### Official Review · Reviewer_q7Qn · 2023-07-08

**Soundness:** 3 good
**Presentation:** 3 good
**Contribution:** 3 good
**Rating:** 6
**Confidence:** 4

**Summary:**

The paper proposes a mask-aware fine-tuning method to address challenges faced by frozen-CLIP-based zero-shot segmentation methods. It addresses the problem of CLIP being insensitive to different mask proposals and tending to produce similar predictions regardless of the variation in proposals. The proposed IP-CLIP successfully assigns appropriate scores to different proposals, unlike the frozen CLIP that exhibits similar scores. Instead of processing each mask individually, the proposed modified CLIP considers all mask proposals simultaneously, thereby reducing computational costs. The experimental results consistently demonstrate that the proposed method outperforms the baselines by a significant margin.

**Strengths:**

 - The proposed method is designed as a plug-and-play approach, making it applicable to any frozen CLIP-based method.
- The proposed method consistently improves the performance of baseline methods, including SegFormer, ZSSeg, and FreeSeg, by substantial margins, particularly on unseen classes.
- The method significantly reduces the computational requirements of CLIP in FreeSeg, and the effectiveness of the proposed mask-aware loss and IP-CLIP is demonstrated through ablation studies.


**Weaknesses:**

- The starting point of the mask attention layer L is determined by a user-defined hyperparameter. The proposed method specifically employs ViT-B/16 as the backbone in the paper. However, if a different backbone is utilized, the selection of this hyperparameter would necessitate a hyperparameter search.

- The notation presented in the paper would be better if it were simplified and clarified.

**Questions:**

- Including experiments with other backbones and proposal generators would enhance the comprehensiveness of the paper


**Limitations:**

The limitations are briefly discussed in the paper, while the societal impact is not addressed.

---

> ### Author Rebuttal · Authors · 2023-08-09
>
> # Response to Reviewer q7Qn
>
> Thank you so much for acknowledging the strength of our method. We have carefully considered your constructive and insightful comments and here are the answers to your concerns.
>
> **Q1. Experiments with other backbones and proposal generators**
>
> To enhance the comprehensiveness of the paper, we extend our experimentation with more Vision-Language Models (CLIP-ViT-L and CLIP-R50) and more proposal generators (SAM-H).
>
> * **Using CLIP-ViT-L for proposal classification**
> We conduct experiments using a stronger Vision-Language Model, CLIP-ViT-L, and the results in cross-dataset setting are shown below. We also include the results of OvSeg with CLIP-ViT-L for comparison. FreeSeg with a standard CLIP-ViT-L model (dubbed `FreeSeg`) still can not achieve satisfactory results. However, by integrating our MAFT (dubbed `FreeSeg+MAFT`), the segmentation results are remarkably enhanced, thus establishing new state-of-the-art benchmarks.
>
> |(ViT-L) | A-847 | A-150 | PC-459 | PC-597 | PAS-20 |
> | -------- | -------- | -------- | -------- | -------- | -------- |
> | OvSeg      | 9.0     | 29.6     |   12.4  | 55.7     | 94.5     |
> | Freeseg    | 8.5     | 21.0     |7.6     | 33.8     | 86.4     |
> | Freeseg + MAFT | 12.1 (+3.6)    | 32.0 (+11.0)    |15.7 (+8.1)   | 58.5 (+25.3)    | 92.1 (+5.7)   |
>
> * **Using CLIP-R50 for proposal classification**
> We also extend our MAFT to CLIP-R50, a Vision-Language Model based on ResNet. Specifically, we modified the $\mathrm{AttentionPool2d}$ unit within CLIP-R50 Image Encoder. The mask proposals are introduced as attention bias ($B$ ) in Multihead Attention, with $F_{cls}$ being repeated N times. Notably in CLIP-R50, $F_{cls}$ is obtained via performing $\mathrm{Global Average Pooling}$ on $F_{feat}$. The results are presented below, the performance on all 5 datasets is improved by a large margin. `Freeseg+MAFT` with CLIP-R50 achieves competitive results with some CLIP-ViT-B-based methods such as OvSeg and OpenSeg, according to Tab. 3 in the paper.
>
> |(R50) | A-847 | A-150 | PC-459 | PC-597 | PAS-20 |
> | -------- | -------- | -------- | -------- | -------- | -------- |
> | Freeseg    |  5.3    | 15.5    |5.4     | 28.2     | 78.1     |
> | Freeseg + MAFT | 8.4 (+3.1)    | 27.0  (+11.5)   |9.9 (+4.5)    | 50.8  (+22.6)    | 89.0  (+10.9)   |
>
>
>
> * **Using Segment Anything Model (SAM-H) as proposal generator**
> Furthermore, we explore using SAM-H as proposal generator. Compared to the Mask2Former trained on COCO-Stuff, SAM-H can produce superior-quality mask proposals. According to **General Response-Q1**, The successful collaboration of SAM-H with our MAFT approach (dubbed `SAM+MAFT`) has yielded remarkable performance enhancements.
>
> In a word, the aforementioned experiments demonstrate the efficacy and robustness of MAFT across different Vision-Language Models and different proposal generators. We will include these experiments in the final version.
>
>
> **Q2. Concerns about defining the hyperparameter $L$**
>
> It is indeed a consideration when utilizing new backbones, the value of $L$ needs to be re-determined. However, it's worth noting that considering the short training duration of our MAFT (as detailed in **Line 221** of the paper), the cost of defining $L$ should be acceptable. To provide further insight, we present the training time for MAFT on different datasets using 4 RTX A6000 GPUs. *e.g.*, on the commonly used COCO-Stuff dataset, MAFT only takes about 20 minutes for training.
>
> ||Traning iterations| Traning times |
> | -------- | -------- | -------- |
> | Pascal-VOC  | 100    | 2 minutes     |
> | COCO-Stuff  | 1000    | 20 minutes     |
> | ADE20K      | 5000     | 100 minutes     |
>
> **Q3. Simplify and clarify the notations**
>
> Thanks for your kind suggestions, we will optimize and clarify the notations in the final version. *e.g.*, $A^c_{IP}$ and $A^c_{fro}$ might be confused with $A^c$. We will simplify them with $A_{S}$ (classification score of the student net) and $A_{T}$ (classification score of the teacher net), while showcasing them in Fig. 2. We believe the simplification and clarification will contribute to the understanding of this work.

---

### Official Review · Reviewer_fJQP · 2023-07-09

**Soundness:** 3 good
**Presentation:** 2 fair
**Contribution:** 2 fair
**Rating:** 5
**Confidence:** 4

**Summary:**

The general goal of the paper is to leverage CLIP for zero-shot segmentation. For this, in contrast to prior work, an CLIP-inspired IP-CLIP encoder is trained to enable mask-level encodings for segmentation. The core of the approach is the so-called IP-encoder that uses as input a frozen mask generator, as well as two losses, a mask-aware loss and a distillation loss

**Strengths:**

The general idea is good and on a high-level the components (IP-CLIP encoder, mask aware loss and CLIP distillation) make sense to me

The reported results are good across three prior baselines and different datasets

**Weaknesses:**

In my view the paper is not well written and important details are either not well motivated or even unclear (see below)

The paper reads to a large extend like an engineering paper with a few changes here and there to adapt to the task as hand. I would assume that to be not so interesting for the majority of NeurIPS

**Questions:**

Here are my main questions about writing

1) A^c is defined to be of dimension NxC (line 119) - with N the number of mask proposals
in the self-distillation loss however, figure 2 is showing a cx1 dimensional vector?
in any case I am confused as standard CLIP (here used as teacher) would not generate a NxC dimensional
map - and thus A^C_{fro} in equation (7) does not seem to make sense to me - can you please explain?

2) related to that: in figure 2 it seems that the final projection from IP-CLIP encoder is used without biases? (w.o. B) - that seems quite obscure to me actually - what is meant?

3) the mask-aware loss seems sensible on a high-level - but the details are not really motivated well. E.g. the reason for normalizing the IoU scores is unclear (equation 5)  - similarly the reason behind the exact formulation of equation 6 remain unclear.

4) the paper uses L layers prior to condition on the masks, and 12-L after that. While there is an experimental ablation about this why would that make sense intuitively?

if the authors can clarify these points I will consider upgrading my score.

detail:
- line 125 "wildly" -> "widely"


post rebuttal
thanks for addressing my questions - I have upgraded mu review as mentioned in my initial review. Please make sure that the improvements are promised are implemented - thanks

**Limitations:**

ok

---

> ### Author Rebuttal · Authors · 2023-08-09
>
> # Response to Reviewer fJQP
>
> Thank you so much for acknowledging the strength of our method. We have carefully considered your constructive and insightful comments and here are the answers to your concerns.
>
> **Q1. Explanation of $A^c \in\mathbb{R}^{C \times N}$, $A^c_{IP}$$\in\mathbb{R}^{C \times 1}$ and $A^{c}_{fro}$$\in\mathbb{R}^{C \times 1}$**
>
> In our framework,  $A^c \in\mathbb{R}^{C \times N}$ is used for computing the *mask-aware loss*. $A^c_{IP} \in \mathbb{R}^{C \times 1}$ and $A^c_{fro} \in \mathbb{R}^{C \times 1}$ are generated only when computing the *self-distillation loss*. Specifically, during the process denoted by the red arrows in Fig. 2, the image propagates into both IP-CLIP and frozen CLIP without incorporating any mask proposals, resulting in  $A^c_{IP}$ and $A^c_{fro}$ respectively. To execute this process at the code level, we invoke the `forward` function of the frozen CLIP and load the parameters of the IP-CLIP to obtain $A^c_{IP}$. Therefore, $A^c_{IP}$ and $A^c_{fro}$ are both vectors of dimensions $C\times 1$. We utilize the $\mathrm{smoothL1}$ in Eq. 7 to enforce equivalence between $A^c_{IP}$ and $A^c_{fro}$.
>
> **Q2. Explanation of the final projection unit and** *w.o.* $B$
>
> First, the final projection unit within CLIP is essentially an MLP module, adept at reshaping the channels of $F_{feat}$ and $F_{cls}$. Notably, the final projection is one component unit in CLIP, and it operates independently of attention bias ($B$).
> Besides, the notation "*w.o.* $B$ " in Fig.2 relates to the process for generating $A^{c}_{IP}$ as described in **Q1**. During this process, the image is directly input to IP-CLIP without incorporating any mask proposal. *i.e.*, this process does not involve attention bias and *repeat* operation.
>
>
> **Q3. Explanation of the *mask-aware loss***
>
> The alignment between $A^c_{select}$ and $S_{IoU}$ is a fundamental goal of the *mask-aware loss*. We observe that in most cases, the maximum value of $A^c_{select}$ tends to approach 1, whereas the maximum value of $S_{IoU}$ ranges from 0.75 to 0.99. This inconsistency can hinder the convergence of the *mask-aware loss*. To address this issue, we introduced a normalization technique for $S_{IoU}$ using Eq. 5. This normalization technique enhances the smoothness of the loss and effectively reduces training variance. Notably, normalization techniques within the loss function have been widely applied in prior works [1*, 2*]. In addition, we experimentally demonstrate the effect of this normalization technique in the subsequent table.
>
> Effect of normalization technique under zero-shot setting (COCO-Stuff):
> | |mIoU$^{s}$ | mIoU$^{u}$ | hIoU| mIoU
> | -------- | -------- | -------- |-------- |- |
> | FreeSeg + MAFT | 43.3     | 50.4     |46.5| 43.9
> | FreeSeg + MAFT (without norm) | 42.1    | 48.7   |45.2| 42.7
>
> Eq. 6 provides the mathematical representation of the $\mathrm{smoothL1}$ function. The primary goal of the proposed *mask-aware loss* is to improve CLIP's classification ability of proposals. Reflecting within the loss function, *i.e.*, aligning $A^c_{select}$ and $S_{IoU}$ by $\mathrm{smoothL1}$.
>
> [1*] Mask Matching Transformer for Few-Shot Segmentation, NeurIPS 2022
>
> [2*] Few-shot semantic segmentation via mask aggregation, arxiv 2022
>
>
> **Q4. Explanation of starting mask attention from layer $L$**
>
> We consider that CLIP's classification significantly relies on context information. If attention bias is introduced right from the first transformer layer, $F_{cls}$ will only interact with the foreground pixels in $F_{feat}$. In this scenario, the loss of context information would be inevitable.
> Therefore, in the initial $L^{th}$ transformer layers of IP-CLIP, we maintain the same design as the original CLIP. $F_{cls}$ utilizes cross-attention with all pixels in $F_{feat}$, thereby retaining the crucial context information. In the subsequent $12-L^{th}$ layers, we introduce the attention bias within MultiHead Attention, with mask-aware fine-tuning to learn the mask-aware representation.
>
> **Q5. Assuming that the paper is not so interesting for the majority of NeurIPS**
>
> It's important to note that zero-shot segmentation constitutes a compelling task within the realm of NeurIPS, as evident from the pertinent works published over recent years [1*, 2*, 3*, 4*].
> We achieve noteworthy performance enhancements in both zero-shot and open-vocabulary settings by appropriate fine-tuning of the CLIP Image Encoder. We emphasize that while our approach is simple, it remains impactful. It introduces a solid and innovative solution that can seamlessly integrate with zero-shot and open-vocabulary methods (*e.g.*, Zegformer, ZSSeg, FreeSeg).
>
> [1*] ReCo: Retrieve and Co-segment for Zero-shot Transfer (NeurIPS 2022)
>
> [2*] The Emergence of Objectness: Learning Zero-shot Segmentation from Videos (NeurIPS 2021)
>
> [3*] Consistent Structural Relation Learning for Zero-Shot Segmentation (NeurIPS 2020)
>
> [4*] Uncertainty-Aware Learning for Zero-Shot Semantic Segmentation (NeurIPS 2020)

---

> > ### Comment · Reviewer_fJQP · 2023-08-18
> >
> > Thanks for the rebuttal. Are the authors planning to update the paper in any way - in particular w.r.t. Q1-Q4? If the authors plan to update, it would be useful to know how (concretely) the authors plan to update the paper.
> >
> > Thanks.

---

> > > ### Author Response · Authors · 2023-08-18
> > > **Response to Reviewer fJQP**
> > >
> > > Thank you again for your valuable comments. We are committed to incorporating the suggested improvements into our paper, referencing **Q1-Q4**. We are confident that these updates will significantly enhance the clarity and overall readability of our work.
> > >
> > > **Q1. Explanation of $A^ c \in\mathbb{R}^{C \times N}$, $A^c_{IP}$$\in\mathbb{R}^{C \times 1}$ and $A^{c}_{fro}$$\in\mathbb{R}^{C \times 1}$**
> > >
> > > * **Simplify and clarify the notations: $A^c_{IP}$ and $A^c_{fro}$**
> > > We notice the potential confusion regarding the notations $A^c_{IP}$, $A^c_{fro}$ and $A^{c}$ for the readers. To address this concern, we have taken steps to enhance clarity.
> > > First, we will introduce additional clarifications at **Line187**: "$A^c_{IP}$ and $A^c_{fro}$ are specifically generated when computing the *self-distillation loss*. It is important to note that when processing an image through IP-CLIP without mask proposals, the resulting $A^c_{IP}$ is a matrix with dimensions ${C \times 1}$."
> > > Besides, we will simplify $A^c_{IP}$ and $A^c_{fro}$ with $A_{S}$ (indicating the student net's score) and $A_{T}$ (indicating the teacher net's score), and showcase them in Fig. 2.
> > >
> > >
> > > **Q2. Explanation of the final projection unit and** *w.o.* $B$
> > >
> > > * **Add explanations of the final projection unit in Fig. 2**
> > > We will enhance the clarity of the final projection unit by adding the following sentence in the caption of Fig. 2: "The final projection unit is an MLP module, used for reshaping the channels of $F_{feat}$ and $F_{cls}$."
> > > * **The notation $w.o.B$  in Fig.2 will be revised as $w.o.M$**
> > > We will revise "*w.o.* $B$ " with "*w.o.* $M$ ", as it corresponds to the process of generating $A^{c}_{IP}$, where the image is input to IP-CLIP without incorporating any mask proposal ($M$). The adjustment to "w.o. $M$ " better conveys the intended meaning and minimizes the potential for confusion.
> > > In addition, the following explanation will be added into the caption of Fig. 2："*w.o.*$M$ denotes IP-CLIP Encoder processes image without utilizing mask proposals ($M$)".
> > >
> > > **Q3. Explanation of the *mask-aware loss***
> > >
> > > * **Clarify the normalization equation in Eq. 5**
> > > At **Line180**, we will update the explanation to："We identify a discrepancy between the maximum values of $A^{c}$ and $S_{IoU}$. The maximum value of $A^{c}$ tends to approach 1, whereas the maximum value of $S_{IoU}$ ranges from 0.75 to 0.99. This inconsistency can hinder the alignment between $A^{c}$ and $S_{IoU}$. Therefore, we introduced a normalization technique for $S_{IoU}$ using Eq. 5."
> > > * **Clarify the $\mathrm{smoothL1}$ function and the *mask-aware loss* in Eq. 6**
> > > Eq. 6 provides the mathematical representation of the $\mathrm{smoothL1}$ function. To make it easier to understand, this equation will be re-formulized as:
> > >
> > > $L_{ma} (A^c_{select}, S_{IoU}^{norm}) = \mathrm{smoothL1} (A^c_{select}, S_{IoU}^{norm}),$
> > > Given two elements $x, y$ in $\mathrm{smoothL1}$. $\mathrm{If} ~ |x - y| < 1$, $\mathrm{smoothL1}(x, y) = 0.5\cdot (x - y)^2$; $\mathrm{Otherwise}$, $\mathrm{smoothL1}(x, y) = |x - y| - 0.5$.
> > >
> > >
> > > **Q4. Explanation of starting mask attention from layer $L$**
> > >
> > > We will refine the sentence beginning at **Line152** as follows: "We consider that CLIP's classification significantly relies on context information. In the first $L$ Transformer layers, the propagation of $F^{i}$ is the same as in standard CLIP. Specifically, $F_{cls}$ utilizes cross-attention with all pixels within $F_{feat}$, effectively retaining the context information."
> > >
> > > ****
> > > We hope our responses have addressed all your concerns. If you have any further questions or comments, please kindly let us know, and we are happy to respond.

---

### Author Rebuttal · Authors · 2023-08-09

# General Response

**Q1. Extending the method with SAM**

As some Reviewers suggest us use stronger proposal generators, *e.g.*, Segment Anything Model (SAM). We explore the performance with SAM-H using an original CLIP (dubbed `SAM+oriCLIP`) or a mask-aware fine-tuned CLIP (dubbed `SAM+MAFT`). In fact, SAM can be seamlessly integrated into our framework as the proposal generator. To provide a comprehensive comparison, we also include the results of `FreeSeg+MAFT`. Experiments are conducted under both zero-shot setting and cross-dataset setting.

Zero-shot setting (Pascal-VOC):
|  |mIoU$^{s}$ | mIoU$^{u}$ | hIoU| mIoU
| -------- | -------- | -------- |-------- |- |
| SAM + oriCLIP | 85.1  | 86.7 | 85.9 | 85.5
| SAM + MAFT | 91.0     | **88.6**    |**89.8**|**90.4**
| FreeSeg + MAFT | **91.4**     | 81.8     |86.3| 89.0

Zero-shot setting (COCO-Stuff):
|  |mIoU$^{s}$ | mIoU$^{u}$ | hIoU| mIoU
| -------- | -------- | -------- |-------- |- |
| SAM + oriCLIP | 43.1    | 43.3   |43.2| 42.1
| SAM + MAFT | **43.4**    | **51.5**   |**47.1**| **44.1**
| FreeSeg + MAFT | 43.3     | 50.4     |46.5| 43.9

Cross-dataset setting:
|  | A-847 | A-150 | PC-459 | PC-597 | PAS-20 |
| -------- | -------- | -------- |-------- |- |- |
| SAM + oriCLIP | 9.0     | 21.3     |7.8     | 33.7      | 87.5    |
| SAM + MAFT | **12.7**     | **33.0**     |**16.2**     | **59.0**      | **92.7**    |
| Freeseg + MAFT | 10.1     | 29.1     |12.8     | 53.5      | 90.0     |


It can be observed that `SAM+MAFT` obtains significant improvement over `SAM+oriCLIP` under both settings. Besides, `SAM+MAFT` also surpasses `Freeseg+MAFT` on all benchmarks. Particularly, in the zero-shot setting (Pascal-VOC), `SAM+MAFT` outperforms `FreeSeg+MAFT` by 6.8\% in terms of mIoU$^{u}$. This enhancement can be attributed to the stronger generalization capabilities of SAM for unseen classes. We will add these results in our final version.

---

### Author Response · Authors · 2023-08-20

Dear Reviewers,

We thank all of you for taking your valuable time to provide insightful comments, which significantly strengthen our paper. We have carefully responded to your questions accordingly with the necessary additional experiments and analyses. We hope our responses have  effectively resolved all your concerns. Since the discussion period is closing soon, if you have any further questions or comments, please let us know, and we are happy to respond.

Paper9803 authors

---

### Decision · Program_Chairs · 2023-09-21

**Decision:**

Accept (poster)

**Comment:**

The paper addresses the problem of zero-shot segmentation by leveraging CLIP via training an encoder to enable mask-level encodings. The proposed approach leverages mask-aware loss and a distillation loss to fine-tune the image-proposals CLIP encoder (IP-CLIP), thereby enabling responses to different mask proposals without significantly degrading the transferrability. The manuscript received ratings of borderline accept,  weak accept, borderline reject and accept. While the reviewers appreciated the idea and effectiveness of the proposed approach, they also raised some concerns such as, writing issues and selection of hyper-parameters. Authors responded to reviewer's queries by submitting a rebuttal. The rebuttal largely addressed the initial concerns raised by the reviewers. Given that three reviewers are generally positive about the paper as well as author's rebuttal, the recommendation is accept. Authors are strongly encouraged to take into account the suggestions of the reviewers as well as changes acknowledged in the rebuttal when preparing the final manuscript.